# DIFFERENCE-BASED GRAPH ATTENTION NETWORKS: A DUAL ATTENTION MECHANISM FOR SIMILARITY AND DISSIMILARITY IN GRAPH LEARNING

## ABSTRACT

Most Graph Neural Networks (GNNs) rely solely on similarity-based attention mechanisms, limiting their ability to distinguish nodes that are structurally similar but semantically distinct. We introduce Difference-Based Graph Attention Network (DGAT), a novel architecture that integrates both similarity and dissimilarity attention within a unified geometric framework. DGAT models contrastive relationships using orthogonal projections and wedge-product approximations, capturing richer feature interactions beyond alignment. Our formulation is grounded in a generalized Iwasawa–Cayley decomposition, where the combination of similarity and dissimilarity attention correspond to orthogonal, scaling, and shifting operations. We also connect its behavior to discrete analogs of differential operators and function orthogonality, establishing a principled geometric interpretation. Experiments across homophilic OGB graphs, specially in OGBl-PPA, and heterophilic benchmarks show that DGAT consistently outperforms GAT, GATv2, and Graph Transformer architectures, especially in settings requiring fine-grained representational contrast or role differentiation.

## 1 INTRODUCTION

Graph-based data structures are ubiquitous in a wide variety of domains such as molecular analysis, social networks, and biological systems. Graph Neural Networks (GNNs) have emerged as powerful models for learning from such data by aggregating and transforming node features through localized neighborhood information. A key advancement in GNNs was the introduction of attention mechanisms, notably in graph attention networks (GATs) (Veličković et al. (2018))(Brody et al. (2022)), which assign different weights to neighbors based on feature similarity. However, traditional attention schemes are limited by their reliance on feature alignment and do not account for dissimilarity or directional contrast between nodes. This limitation reduces their expressivity, especially in settings involving role differentiation, heterophily (Luan et al. (2022), or structural asymmetries.

In this paper, we propose Difference-Based Graph Attention Network(DGAT), a novel architecture that extends the attention paradigm to simultaneously capture similarity and dissimilarity in node interactions. DGAT introduces a dual attention mechanism where similarity is captured through additive or cosine attention, while dissimilarity is modeled via angular deviation using orthogonal projections and wedge product approximations. This dual-path formulation mitigates common issues such as oversmoothing (Wu et al. (2023)) and embedding collapse in structurally symmetric graphs, while preserving geometric interpretability.

Furthermore, DGAT is designed with a modular structure inspired by the Iwasawa decomposition in Lie theory, aligning attention components with orthogonal, scaling, and shifting operations. We provide theoretical guarantees showing that DGAT is a non-expansive and convergent operator, with bounded error propagation. Our contributions are as follows:

- We introduce a principled attention mechanism that integrates similarity and dissimilarity signals using orthogonal projections and geometric algebra.
- We provide a theoretical analysis demonstrating the boundedness and non-expansiveness of DGAT, with convergence guarantees under standard assumptions.

- We evaluate DGAT across node classification, graph classification, and link prediction tasks on multiple benchmarks both homophilic and heterophilic, showing consistent improvements over GAT, GATv2, and Graph Transformer baselines.

The rest of the paper is organized as follows: Section 2 reviews related work on attention-based GNNs and prior efforts in geometric modeling. Section 3 presents DGAT architecture and its theoretical foundation. Section 4 describes the experimental setup. Section 5 reports results and analysis. Section 6 discusses implications and extensions. Section 7 concludes with future directions.

## 2 RELATED WORK

The literature on Graph Neural Networks has evolved from shallow neighborhood-averaging (Hamilton et al. (2018)) models to highly expressive architectures incorporating message passing, attention, and geometric priors. The Graph Convolutional Network (GCN) (Kipf & Welling (2016)) introduced spectral filtering with neighborhood aggregation. GAT extended this by introducing attention coefficients over node pairs, assigning learned weights based on feature similarity. GATv2 improved this mechanism by simplifying and generalizing attention computation to remove static patterns.

More recently, Graph Transformer models have drawn inspiration from sequence models, leveraging global self-attention, edge encodings, and learnable positional biases to enable long-range information propagation (Cai & Lam (2019)). These models incorporate relative or structural encoding into the attention layers, often achieving state-of-the-art performance in node and graph-level tasks. Despite these advances, nearly all attention (Vaswani et al. (2017)) mechanisms in GNNs rely exclusively on similarity (dot product, additive forms), implicitly treating dissimilar neighbors as noise.

This assumption has limited the capacity of GNNs to distinguish semantically distinct but structurally aligned nodes, which is especially problematic in role-differentiated graphs or heterophilic settings. While efforts such as Orthogonal GNNs (Guo et al. (2021)) have proposed architectural modifications to preserve orthogonality (Kondor et al. (2018)) or reduce feature collapse, they do not explicitly model dissimilarity as a primary signal in attention.

Our work addresses this gap by introducing a dissimilarity-aware attention (Majhi et al. (2021))(Lee et al. (2023a)) formulation. We incorporate geometric tools (orthogonal projections, wedge products, and directional contrasts) into the attention mechanism itself. By pairing these with learnable gating and combining them with classical similarity-based scores, DGAT offers a balanced and expressive model that can adaptively weigh alignment and contrast across graph neighborhoods. This contributes a novel inductive bias that can benefit both homophilic and heterophilic settings, offering enhanced flexibility and interpretability

## 3 METHODOLOGY

### 3.1 DGAT ARCHITECTURE

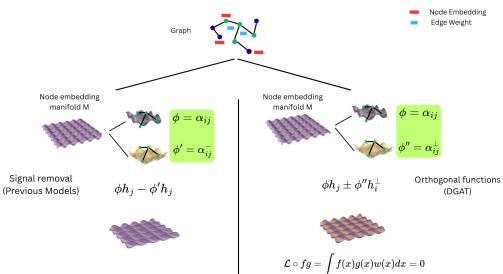

Figure 1: Graphical description of the DGAT approach (right) for handling low- and high-frequency signals, highlighting its differences from previous models (left).

DGAT (Figure 1) builds upon the standard message-passing framework, where node representations are updated through attention-weighted aggregation of neighbors. The key novelty in DGAT is the incorporation of two parallel attention components:

- A **similarity attention** path using cosine similarity or additive attention (equation 1).
- A **dissimilarity attention** path using angular contrast via functional orthogonal projection or vector difference (equation 2).

$$h_i^{sim} = \sum_{j \in \mathcal{N}(i)} \alpha_{ij}^{sim} * h_j \tag{1}$$

$$h_i^{dis} = \sum_{j \in \mathcal{N}(i)} (\alpha_{ij}^{dis} * \alpha_{ij}^{sim}) * \begin{cases} \lambda * (h_i - \frac{\langle h_i, h_j \rangle}{\langle h_j, h_j \rangle + \epsilon} * h_j) & \text{if } inner - orth \\ h_i & otherwise \end{cases} \tag{2}$$

The final representation of layer $l + 1$ of node $i$ obtained from the representation of layer $l$, is computed as a gated combination of the two components (equation 3):

$$h_i = h_i^{sim} + g(h_i^{dis}) \tag{3}$$

Here, the terms of these equations are defined as follows:

- $\alpha_{ij}^{\text{sim}}$: Similarity-based attention coefficient (e.g., dot product or additive mechanism).

- $\alpha_{ij}^{\text{dis}}$: Dissimilarity-based attention coefficient, measuring angular or orthogonal deviation. It acts as an implicit attention mechanism for the gating function. During training, dropout perturbs the softmax-normalized coefficients, allowing the attention to adaptively modulate both the gate and the node parameters, learning how dissimilarity contributes to a node's representation. At inference, even if the inner orthogonal component is inactive, the gate preserves the node's embedding, ensuring a stable self-contribution. The term $\langle h_i, h_j \rangle$ denotes the inner product between the node embeddings $h_i$ and $h_j$. A learned scalar $\lambda \in \mathbb{R}$ serves as a scaling coefficient and the $\epsilon$ is a small integer that avoid division by 0. The equation offers the possibility to add a function to $h_j$. One way to introduce such a function is by combining DGAT with the mechanism proposed in GARN Wang et al. (2024). This would enable the unified modeling of similarity (GAT: $cos\alpha$), opposite of similarity (GARN: $-cos\alpha$), and orthogonality (DGAT: $w|sin\alpha|$). The function applied to $h_j$ could be linear, nonlinear, or even discontinuous. However, for the purposes of this work, we have not applied any additional transformation to $h_j$. Although our focus is on the orthogonal component, the full formulation could also incorporate the $-cos\alpha$. Approaches such as A2GCN (Ai et al. (2024)) might provide a complementary perspective for this component, but doing so would require an additional correction term involving $I - k_i L$ together with $\alpha^{dis}$ to ensure geometric consistency. Such a correction would control the interaction between frequency-domain filtering and embedding-space orthogonality, enabling more stable multi-layer propagation. The gates' learned parameters induce an effect analogous to the $k_i$ parameter in A2GCN, but without explicitly orthogonalizing the Laplacian matrix.

- $g$: A learned gating function (Note it always returns a vector that regulates the contribution of each component). Two optional variants are considered: (i) an orthogonality-based gate that enforces function orthogonality, that we named $w$ orthogonal, between the similarity and dissimilarity components across all nodes, effectively treating similarity embeddings as globally functional orthogonal to their dissimilarity counterparts; and (ii) a Lie bracket–inspired gate that models torsion in combination with the orthogonality gate. In this work, the torsion gate is implemented with a ReLU nonlinearity, although other activation functions (or even omitting it) could be explored depending on the problem setting.

$$g(h_i^{dis}) = \begin{cases} \lambda * (\boldsymbol{h}^{dis} - \frac{\langle \boldsymbol{h}^{dis}, \boldsymbol{h}^{sim} \rangle}{\langle \boldsymbol{h}^{sim}, \boldsymbol{h}^{sim} \rangle + \epsilon} * \boldsymbol{w}_\theta * h^{sim}) & \text{if } \boldsymbol{w} - orth \\ ReLU(\boldsymbol{h}_i^{sim} - \boldsymbol{W} * \boldsymbol{h}_i^{dis}) * h_i^{dis} & \text{if } torsion \\ h_i^{dis} & otherwise \end{cases} \tag{4}$$

- $\mathcal{N}(i)$: 1-step Neighborhood of node i.

The combination of similarity and dissimilarity reflects both symmetric and antisymmetric contributions to the message. The symmetric term (mean) reinforces shared structure, while the antisymmetric term (difference) introduces directional variation, enabling contrastive feature propagation even among nodes with similar initial embeddings.

We provide two variants of DGAT: one based on GAT-style additive attention (Veličković et al. (2018); Brody et al. (2022)), and another inspired by Graph Transformer mechanisms.

**GAT-style attention (additive form):**

$$\alpha_{ij}^{sim} = \text{softmax}\left(\boldsymbol{W}(\boldsymbol{h}_i + \boldsymbol{h}_j + \boldsymbol{e}_{ij})\right) \tag{5}$$

$$\alpha_{dis}^{gram} = \boldsymbol{h}_i - \frac{\langle \boldsymbol{h}_i, \boldsymbol{h}_j + \boldsymbol{e}_{ij} \rangle}{\langle \boldsymbol{h}_j + \boldsymbol{e}_{ij}, \boldsymbol{h}_j + \boldsymbol{e}_{ij} \rangle + \epsilon} * (\boldsymbol{h}_j + \boldsymbol{e}_{ij})$$

$$\alpha_{ij}^{dis} = \begin{cases} dropout(\text{softmax}\left(\lambda_{dis} * \boldsymbol{W} * \alpha_{dis}^{gram}\right)) & if\ training \\ \text{softmax}\left(\lambda_{dis} * \boldsymbol{W} * \alpha_{dis}^{gram}\right) & otherwise \end{cases} \tag{6}$$

This formulation enables DGAT to contrast direct similarity with orthogonal deviation. Although $1-\cos(\theta)$ and $\sin(\theta)$ are not identical, they grow similarly with angular difference. For precision, we instead compute the orthogonal component via a Gram projection, subtracting from $h_i$ its projection onto $(h_j + e_{ij})$.

**Transformer-style attention:**  To demonstrate DGAT's generality, we also implement a variant with Graph Transformer-style attention, replacing additive attention with queries and keys:

$$\alpha_{ij}^{sim} = dropout(\text{softmax}\left(\boldsymbol{Q}^T(\boldsymbol{K} + \boldsymbol{W}_e^{sim} * \boldsymbol{e}_{ij})\right) \tag{7}$$

$$\alpha_{ij}^{dis} = \begin{cases} dropout(\text{softmax}\left(\|\boldsymbol{Q}^T \wedge (\boldsymbol{K} + \boldsymbol{W}_e^{dis} * \boldsymbol{e}_{ij})\|\right)) & if\ training \\ \text{softmax}\left(\|\boldsymbol{Q}^T \wedge (\boldsymbol{K} + \boldsymbol{W}_e^{dis} * \boldsymbol{e}_{ij})\|\right) & otherwise \end{cases}$$

$$\|\boldsymbol{Q}^T \wedge_{ij}^{dis(trans)} (\boldsymbol{K} + \boldsymbol{W}_e^{dis} * \boldsymbol{e}_{ij})\| = \|\boldsymbol{Q}\| * \|\boldsymbol{K}\| - \langle \boldsymbol{Q}, \boldsymbol{K} + \boldsymbol{W}_e * \boldsymbol{e}_{ij} \rangle$$

$$\|\boldsymbol{Q}^T \wedge_{ij}^{dis(gram)} (\boldsymbol{K} + \boldsymbol{W}_e^{dis} * \boldsymbol{e}_{ij})\| = \lambda_{dis} * (\boldsymbol{Q} - \frac{\langle \boldsymbol{Q}, \boldsymbol{K} + \boldsymbol{W}_e * \boldsymbol{e}_{ij} \rangle}{\langle \boldsymbol{K} + \boldsymbol{W}_e * \boldsymbol{e}_{ij}, \boldsymbol{K} + \boldsymbol{W}_e * \boldsymbol{e}_{ij} \rangle + \epsilon} * (\boldsymbol{K} + \boldsymbol{W}_e * \boldsymbol{e}_{ij})) \tag{8}$$

In this formulation, the wedge product ($\wedge$) captures the geometric component orthogonal to the direction of alignment, effectively substituting the classical matrix multiplication with a formulation based on the norm of the wedge product. This provides access to orthogonal information, allowing the model to attend to latent contrast that is often missed by dot-product-based attention. We note that a variant could modify the Q or K vectors to simulate curvature changes in the orthogonal component, with a scaling parameter applied to preserve the norm and maintain the curvature effect. The mathematical details of the orthogonality and the implementation of DGAT can be found in the appendix section A.2.

## 4 EXPERIMENTS

### 4.1 DATASETS

We evaluate DGAT on four widely used benchmark datasets from the Open Graph Benchmark (OGB) (Hu et al. (2020)) suite, each targeting a different type of graph learning task:

- **OGBg-MolHIV**: A graph-level binary classification dataset composed of molecular graphs. The task is to predict whether a molecule inhibits HIV replication.

- **OGBn-Proteins**: A node classification dataset constructed from protein interaction networks. Each node represents a protein, and the task is to predict protein functions based on structural context.

- **OGBl-PPA**: A link prediction dataset based on protein-protein association graphs. The goal is to predict missing interactions between proteins.

- **OGBl-DDI**: A link prediction dataset based on drug-drug interaction graphs. The goal is to predict missing interactions between drugs.

These datasets were selected to evaluate DGAT performance across the three major graph learning problems: graph classification, node classification, and link prediction. All datasets are provided through the `ogb` Python package and are fully compatible with PyTorch Geometric, enabling standardized preprocessing and evaluation protocols.

Each dataset defines its own evaluation metric: OGBg-MolHIV and OGBn-Proteins use the Area Under the Receiver Operating Characteristic curve (AUROC) on the test set, while OGBl-PPA and OGBl-DDI are evaluated using the Hits@100 metric and Hits@20 respectively. These metrics align with the established benchmarks for the respective tasks and enable consistent comparison with prior work.

In addition to covering diverse task types, these datasets pose unique structural and semantic challenges. OGBg-MolHIV contains many molecules with similar topologies but different biological activity, requiring fine-grained attention. OGBn-Proteins features sparse, binary node attributes and complex relational patterns, testing the model's sensitivity to local structure. OGBl-PPA represents a very large and sparse graph, where long-range dependencies and efficient dissimilarity modeling are essential. OGBl-DDI consists of small unweighted graphs; since trainable features are often noisy, incorporating geometric information enhances expressivity and provides an additional layer of structure.

For heterophilic graphs, we considered four benchmarks: **Minesweeper**, a synthetic binary classification dataset where nodes represent cells of the game and the task is to predict whether a cell contains a bomb; **Roman-Empire**, a node classification dataset constructed from the corresponding Wikipedia page where words are classified based on structural context; **Amazon-Ratings**, a node prediction dataset derived from Amazon-Ratings data with the goal of predicting a node's rating from its neighbors; and **Questions**, a node prediction dataset built from questionnaire responses where the objective is to infer answers from neighboring nodes. Performance is evaluated using AUROC for Minesweeper and Questions, and accuracy for Roman-Empire and Amazon-Ratings (Platonov et al. (2024)).

## 4.2 BASELINES

To evaluate the effectiveness of DGAT, we compared it against a set of established Graph Neural Network (GNN) architectures that incorporate attention mechanisms, which allow for a fair assessment of the benefit of introducing dissimilarity-aware mechanisms. The following main baselines were selected because they capture the evolution of attention design in graph learning and cover both localized and global attention paradigms.

- **GAT** (Veličković et al. (2018)): The Graph Attention Network introduces node-level attention into message passing. It computes edge-wise attention scores using additive transformations of node features and aggregates messages from neighbors with these learned weights. GAT serves as a foundational model for attention-based GNNs.

- **GATv2** (Brody et al. (2022)): This variant improves upon GAT by enabling dynamic attention scores that depend symmetrically on both source and target node features. It eliminates the static transformation in the original GAT and provides a more expressive and flexible attention mechanism.

- **Graph Transformer (GTransf)** (Shi et al. (2021); Yun et al. (2019)): GTransf adapts transformer-style global attention to the graph domain. It uses pairwise queries and keys, enhanced by relative positional encodings or structural priors. Additionally, it incorporates TransE-style edge embeddings, making it more effective in capturing long-range and relational dependencies.

These models were chosen not only for their performance on standard benchmarks, but also for their diverse attention formulations—ranging from local additive (GAT), to symmetric dynamic (GATv2), to global pairwise (Graph Transformer). This diversity allows us to fairly evaluate the benefit of introducing dissimilarity-aware mechanisms across different attention regimes. In particular, GAT and GATv2 enable comparison within additive frameworks, where DGAT's dissimilarity attention can be directly substituted for similarity attention. The Graph Transformer comparison highlights the generalizability of DGAT across architectures, especially in more complex, transductive settings.

In addition, for completeness, we also included a set of commonly used baselines. We selected the classical models GCN (Kipf & Welling (2016)), GraphSage (Hamilton et al. (2018)), and FAGCN (Bo et al. (2021)), which operate without attention mechanisms. Moreover, based on GAT, we have added Polynormer (Deng et al. (2024)) and GARN (Wang et al. (2024)) that extend GAT to model more complex scenarios. GARN is also based in FAGCN combining high and low signals with attention for dissimilarity removal. The main difference between GARN and DGAT is that our approach extracts information from the dissimilarity rather than discarding the undesired component. Finally, based on graph transformers we have considered Exphormer (Shirzad et al. (2023)).

## 4.3 EXPERIMENTAL SETTINGS

To ensure a fair comparison, all models—including DGAT and the baselines—were implemented using the same architectural configurations, differing only in the choice of convolutional layer. The architectural configuration used for homophilic and heterophilic graphs differs, but is preserved across all models and baselines. Each model consists of multiple attention layers interleaved with batch normalization and dropout, with the final layer using batch normalization only. The architecture for homophilic graphs is illustrated in Figure 4 in the appendix.

To compensate for the extra parameters from DGAT's dual-attention mechanism, we increased the number of attention heads in baseline models to match the parameter count, typically by a factor of two (except in OGBn-Proteins). This ensured that DGAT's gains were not due to larger capacity but to dissimilarity-based attention. All DGAT variants (DGAT-Transformer, DGAT-Transformer-Gram, DGATv2) used identical hyperparameters, and their optimal settings were consistent across versions.

Importantly, we do not replicate the leaderboard-scale architectures (Luo et al. (2024)) reported in the OGB benchmarks. Instead, we adopt a controlled mid-scale configuration that allows consistent testing across all models on consumer hardware (an AMD Radeon RX 7900 GRE with 16GB VRAM for OGBg-MolHIV, OGBl-DDI, Roman-Empire, and Minesweeper. NVIDIA GeForce RTX 3090 Ti with 32GB VRAM for OGBn-Proteins, OGBl-PPA, Amazon-Ratings and Questions. Both setups were complemented with an Intel i7 CPU). This decision ensures that DGAT can be evaluated fairly under practical deployment conditions. As a result, our absolute scores may differ from those reported on the OGB leaderboard and papers, but our focus is on the **relative improvement** over strong baselines under **identical architectural and resource constraints**.

The hyperparameter tested were as follows: starting learning rate values of 0.03, 0.01, and 0.005; weight decay values of 1e-8, 0.00001, and 0.00005; and dropout values of 0.1, 0.3, and 0.5. The number of training epochs was fixed at 100 for OGBg-MolHIV, 1000 for OGBn-Proteins, and 100 for OGBl-PPA. In addition, a learning-rate scheduler was used to fully test our model. Specifically, we used the StepLR scheduler. The final hyperparameter configurations for each dataset and model are shown in Tables 3 to 12 in the appendix.

Hyperparameters such as learning rate, weight decay, dropout rate, and number of layers were tuned on the validation set using grid search for each model. Training was conducted with early stopping based on validation performance to prevent overfitting.

This experimental protocol emphasizes both fairness and reproducibility, allowing us to isolate the contribution of dissimilarity-aware attention in DGAT.

## 4.4 RESULTS AND ANALYSIS

For each dataset we obtained the results of DGAT using both versions of the attention, based in GATv2 and based in GraphTransformer, and we recalculated the results for all baselines.

To ensure fairness, all models were trained under the same conditions. For homophilic graphs, the experiments were performed using the architecture presented in Figure 4 in the appendix. For heterophilic graphs, we adopted the architecture proposed in previously published GNN works with public available code, ensuring comparable performance to the results reported by those works.

DGAT consistently outperforms the baselines across most tasks. Notably, DGAT achieves significant improvements on OGBg-MolHIV, OGBl-PPA and OGBl-DDI, where structurally similar molecules may exhibit different biological activities. The dissimilarity-based attention component enhances role differentiation and mitigates oversmoothing in densely connected regions.

It is important to highlight that, in the worst case, DGAT reduces to GATv2 or Graph Transformer depending on the implementation of the difference mechanism. By explicitly incorporating dissimilarity, DGAT not only complements equivariant information but also gains flexibility to overcome the limitations of solely dot-product-based attention.

The final results with the selected parameters are shown in Table 1 and Table 2. Each entry reports the maximum test value across runs, with the standard deviation representing the difference between best results. Color coding indicates best model of each type: green for GAT variants and blue for Graph Transformer variants, with darker shades highlighting the best result per dataset.

To illustrate the efficiency of the head compensation strategy and the resulting parameter counts, we provide a comparison in Figure 2. This highlights how the adjustment balances the potential parameter increase of DGAT relative to the baselines.

Table 1: Validation and test performance on OGB benchmarks. OGBg-MolHIV and OGBn-Proteins report **maximum** AUROC; OGBl-PPA reports **maximum** Hits@100; OGBl-DDI reports **maximum** Hits@20. Best results per attention variant are highlighted (green: GAT-based layers; blue: transformer-based layers). Hyperparameters are listed in Tables 3 to 12 in the appendix.

| Model | MolHIV | | Proteins | |
|---|---|---|---|---|
| | Val | Test | Val | Test |
| GCN | $0.6063_{\pm.0060}$ | $0.6743_{\pm.0031}$ | $0.7701_{\pm.0014}$ | $0.6980_{\pm.0017}$ |
| GraphSage | $0.6581_{\pm.0040}$ | $0.6868_{\pm.0055}$ | $0.7969_{\pm.0015}$ | $0.7574_{\pm.0040}$ |
| FAGCN | $0.5371_{\pm.0029}$ | $0.6993_{\pm.0011}$ | $\mathbf{0.8372}_{\pm.0015}$ | $\mathbf{0.7833}_{\pm.0025}$ |
| GAT | $0.6970_{\pm.0040}$ | $0.6980_{\pm.0070}$ | $0.6974_{\pm.0020}$ | $0.6997_{\pm.0030}$ |
| GATv2 | $0.6917_{\pm.0040}$ | $0.7137_{\pm.0040}$ | $0.8203_{\pm.0010}$ | $0.7778_{\pm.0036}$ |
| Polynormer | $0.6985_{\pm.0030}$ | $0.7186_{\pm.0032}$ | $0.7727_{\pm.0031}$ | $0.7132_{\pm.0042}$ |
| GTransf | $0.6921_{\pm.0060}$ | $0.7123_{\pm.0050}$ | $0.8145_{\pm.0020}$ | $0.7783_{\pm.0010}$ |
| Exphormer | $0.6598_{\pm.0042}$ | $0.6707_{\pm.0058}$ | $0.7778_{\pm.0021}$ | $0.7441_{\pm.0038}$ |
| GARN | $0.6555_{\pm.0015}$ | $0.7197_{\pm.0010}$ | $0.8181_{\pm.0018}$ | $0.7744_{\pm.0039}$ |
| DGATv2 | $\mathbf{0.6982}_{\pm.0039}$ | $\mathbf{0.7305}_{\pm.0005}$ | $\mathbf{0.8255}_{\pm.0020}$ | $\mathbf{0.7810}_{\pm.0010}$ |
| DGAT-Transf | $\mathbf{0.7196}_{\pm.0110}$ | $\mathbf{0.7361}_{\pm.0050}$ | $\mathbf{0.8260}_{\pm.0020}$ | $\mathbf{0.7834}_{\pm.0010}$ |
| DGAT-Transf-Gram | $\mathbf{0.7360}_{\pm.0070}$ | $\mathbf{0.7438}_{\pm.0050}$ | $\mathbf{0.8235}_{\pm.0010}$ | $\mathbf{0.7830}_{\pm.0020}$ |
| **Model** | **PPA** | | **DDI** | |
| | Val | Test | Val | Test |
| GCN | $0.0838_{\pm.0241}$ | $0.0896_{\pm.0230}$ | $0.4727_{\pm.0103}$ | $0.3671_{\pm.0324}$ |
| GraphSage | $0.0593_{\pm.0156}$ | $0.0608_{\pm.0131}$ | $0.6218_{\pm.0019}$ | $0.4439_{\pm.0175}$ |
| FAGCN | $0.0000_{\pm.0000}$ | $0.0000_{\pm.0000}$ | $0.6321_{\pm.0015}$ | $0.3018_{\pm.0025}$ |
| GAT | $0.3380_{\pm.0408}$ | $0.3421_{\pm.0400}$ | $\mathbf{0.5539}_{\pm.0022}$ | $\mathbf{0.4114}_{\pm.0188}$ |
| GATv2 | $0.8964_{\pm.0130}$ | $0.8961_{\pm.0120}$ | $0.4597_{\pm.0246}$ | $0.2595_{\pm.0135}$ |
| Polynormer | $0.2234_{\pm.0230}$ | $0.2235_{\pm.0228}$ | $0.4804_{\pm.0083}$ | $0.2286_{\pm.0065}$ |
| GTransf | $0.1138_{\pm.1231}$ | $0.1327_{\pm.1250}$ | $\mathbf{0.6369}_{\pm.0020}$ | $0.4702_{\pm.0586}$ |
| Exphormer | $0.0385_{\pm.0133}$ | $0.0386_{\pm.0120}$ | $0.5701_{\pm.0023}$ | $0.3321_{\pm.0003}$ |
| GARN | $\mathbf{0.9743}_{\pm.0055}$ | $\mathbf{0.9760}_{\pm.0041}$ | $0.6348_{\pm.0048}$ | $\mathbf{0.4640}_{\pm.0313}$ |
| DGATv2 | $\mathbf{0.9838}_{\pm.0050}$ | $\mathbf{0.9865}_{\pm.0047}$ | $0.6088_{\pm.0052}$ | $\mathbf{0.5026}_{\pm.0130}$ |
| DGAT-Transf | $\mathbf{0.1346}_{\pm.0227}$ | $\mathbf{0.1346}_{\pm.0230}$ | $0.5640_{\pm.0027}$ | $0.1023_{\pm.0009}$ |
| DGAT-Transf-Gram | $\mathbf{0.1976}_{\pm.0288}$ | $\mathbf{0.1979}_{\pm.0314}$ | $\mathbf{0.5938}_{\pm.0179}$ | $\mathbf{0.4748}_{\pm.0112}$ |

Table 2: Validation and test performance on heterophilic benchmarks for a small amount of layers. Roman-Empire and Amazon-Ratings report the **maximum** accuracy while Minesweeper and Questions report **maximum** AUROC. Best results per attention variant are highlighted (green: GAT-based layers; blue: transformer-based layers). Hyperparameters are listed in Table 13 in the appendix. All models use a JKNet and ResNet for the comparison. (Values marked with * were obtained using Polynormer global attention replacing GAT with DGATv2; otherwise the test value is 0.9172)

| Model | Minesweeper | | Roman-Empire | |
|---|---|---|---|---|
| | Val | Test | Val | Test |
| GCN | $0.8923_{\pm.0053}$ | $0.8960_{\pm.0052}$ | $0.9084_{\pm.0015}$ | $0.9080_{\pm.0007}$ |
| GraphSage | $0.8946_{\pm.0119}$ | $0.8980_{\pm.0030}$ | $0.9029_{\pm.0081}$ | $0.9045_{\pm.0033}$ |
| FAGCN | $0.8628_{\pm.0092}$ | $0.8630_{\pm.0011}$ | $0.8492_{\pm.0125}$ | $0.8378_{\pm.0002}$ |
| GAT | $0.8994_{\pm.0014}$ | $0.9102_{\pm.0016}$ | $0.9157_{\pm.0034}$ | $0.8980_{\pm.0038}$ |
| GATv2 | $\textbf{0.9032}_{\pm.0040}$ | $\textbf{0.9140}_{\pm.0082}$ | $0.8959_{\pm.0039}$ | $0.8948_{\pm.0042}$ |
| Polynormer | $0.8933_{\pm.0073}$ | $0.8964_{\pm.0070}$ | $\textbf{0.9201}_{\pm.0030}$ | $\textbf{0.9223}_{\pm.0032}$ |
| GTransf | $0.8914_{\pm.0081}$ | $0.8987_{\pm.0070}$ | $0.8902_{\pm.0041}$ | $0.8872_{\pm.0034}$ |
| Exphormer | $0.9024_{\pm.0051}$ | $0.9086_{\pm.0083}$ | $0.8990_{\pm.0015}$ | $0.8994_{\pm.0017}$ |
| GARN | $0.8552_{\pm.0064}$ | $0.8642_{\pm.0072}$ | $0.9076_{\pm.0012}$ | $0.9068_{\pm.0020}$ |
| DGATv2 | $\textbf{0.9077}_{\pm.0089}$ | $\textbf{0.9148}_{\pm.0084}$ | $\textbf{0.9220*}_{\pm.0038}$ | $\textbf{0.9257*}_{\pm.0040}$ |
| DGAT-Transf | $\textbf{0.8970}_{\pm.0035}$ | $\textbf{0.9092}_{\pm.0034}$ | $\textbf{0.9091}_{\pm.0033}$ | $\textbf{0.9056}_{\pm.0037}$ |
| DGAT-Transf-Gram | $\textbf{0.9015}_{\pm.0097}$ | $\textbf{0.9095}_{\pm.0074}$ | $\textbf{0.9043}_{\pm.0037}$ | $\textbf{0.9010}_{\pm.0089}$ |
| **Model** | **Amazon-Ratings** | | **Questions** | |
| | Val | Test | Val | Test |
| GCN | $0.5378_{\pm.0060}$ | $0.5439_{\pm.0075}$ | $\textbf{0.9729}_{\pm.0011}$ | $\textbf{0.9724}_{\pm.0010}$ |
| GraphSage | $0.5532_{\pm.0029}$ | $0.5578_{\pm.0028}$ | $\textbf{0.9728}_{\pm.0010}$ | $\textbf{0.9724}_{\pm.0008}$ |
| FAGCN | $0.5172_{\pm.0082}$ | $0.5288_{\pm.0091}$ | $0.9725_{\pm.0009}$ | $0.9716_{\pm.0009}$ |
| GAT | $0.5490_{\pm.0043}$ | $0.5550_{\pm.0049}$ | $0.9720_{\pm.0008}$ | $0.9723_{\pm.0009}$ |
| GATv2 | $0.5491_{\pm.0064}$ | $0.5520_{\pm.0067}$ | $\textbf{0.9720}_{\pm.0008}$ | $\textbf{0.9725}_{\pm.0009}$ |
| Polynormer | $0.5339_{\pm.0031}$ | $0.5480_{\pm.0066}$ | $0.9727_{\pm.0008}$ | $0.9715_{\pm.0009}$ |
| GTransf | $\textbf{0.5485}_{\pm.0048}$ | $\textbf{0.5590}_{\pm.0052}$ | $0.9722_{\pm.0008}$ | $0.9712_{\pm.0009}$ |
| Exphormer | $0.5512_{\pm.0042}$ | $0.5552_{\pm.0058}$ | $0.9722_{\pm.0009}$ | $0.9723_{\pm.0009}$ |
| GARN | $0.4793_{\pm.0058}$ | $0.4853_{\pm.0059}$ | $0.9724_{\pm.0009}$ | $0.9720_{\pm.0009}$ |
| DGATv2 | $\textbf{0.5515}_{\pm.0064}$ | $\textbf{0.5581}_{\pm.0064}$ | $\textbf{0.9728}_{\pm.0009}$ | $\textbf{0.9729}_{\pm.0009}$ |
| DGAT-Transf | $\textbf{0.5492}_{\pm.0048}$ | $\textbf{0.5582}_{\pm.0055}$ | $0.9722_{\pm.0008}$ | $0.9723_{\pm.0009}$ |
| DGAT-Transf-Gram | $\textbf{0.5492}_{\pm.0056}$ | $\textbf{0.5580}_{\pm.0064}$ | $\textbf{0.9723}_{\pm.0008}$ | $\textbf{0.9725}_{\pm.0009}$ |

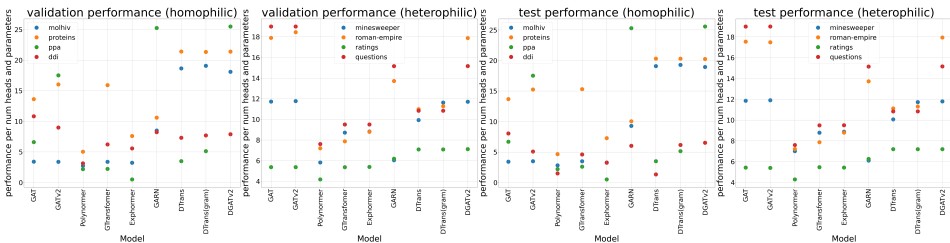

Figure 2: Comparison of model efficiency with respect to the number of parameters and attention heads across models. For clarity of visualization, the values corresponding to the Graph Transformer variants have been scaled by a factor of 100, due to the substantial parameter count required by these models.

## 5 DISCUSSION

DGAT provides a modular and geometry-aware attention mechanism that extends beyond traditional similarity-based models (Li et al. (2024)). Its architecture is inherently flexible, supporting any pair of similarity and dissimilarity functions. While this paper primarily uses cosine similarity and orthogonal projection, the framework can generalize to other metrics, such as structural similarity, Hamming distance, or symbolic dissimilarity, making it suitable for a wide range of graph structures and tasks.

From a geometric perspective, DGAT operates not only over aligned vector components (as in conventional attention), but also over orthogonal or anti-aligned directions. This dual representation enables richer modeling of node relationships, particularly in graphs where structural symmetry or homophily conceals functional differences. By measuring deviation rather than alignment, the dissimilarity component introduces a complementary inductive bias that is especially beneficial for identifying node roles, community boundaries, and structural anomalies.

DGAT's design also draws inspiration from classical mathematical constructs, particularly the Iwasawa decomposition of Lie groups. In this analogy, similarity attention plays the role of the orthogonal group $K$ in Archimedean geometries, while the combination of similarity and dissimilarity attention mirrors the scaling matrix $A$. The gating mechanism corresponds to the unipotent shift $N$, completing the decomposition. This interpretation situates DGAT within a broader theoretical framework, bridging connections to group theory, identity-preserving symmetric spaces (Zhang et al. (2023)), and potentially equivariant neural architectures (Batzner et al. (2022)).

The capacity of DGAT to operate over both similarity and dissimilarity also introduces the potential to improve the learning of anti-symmetric or direction-sensitive attention mechanisms (such as those based on wedge or cross products). These geometric formulations are particularly promising in tasks involving directed graphs, temporal sequences, or causal reasoning. Additionally, DGAT directly addresses the problem of representational collapse in symmetric graph regions, by embedding angular contrast directly at the attention level.

DGAT also shows promise in heterophilic and relational graphs (Zhu et al. (2024)), where the assumption of feature similarity between connected nodes does not hold. In such settings, the dissimilarity pathway enables flexible message propagation while preserving the identity of semantically distinct nodes, making the model suitable for tasks such as fraud detection, role classification, and learning over knowledge graphs with mixed semantics.

Our empirical results demonstrate that this geometric approach particularly excels in noisy settings and on graphs with mixed homophily, where is essential to preserve the correlation between nodes.

Similar motivations have appeared in other recent works such as (Wang et al. (2024)), which introduce sign corrections in attention to handle heterophily. However, such approaches primarily address directional inversion and do not fully capture the deeper geometric duality between similarity and dissimilarity. DGAT addresses this by explicitly modeling two complementary views (similarity and dissimilarity) under a unified, gated architecture (Li et al. (2017)), leading to a richer and more expressive attention mechanism. In scenarios where the orthogonal component is fully noisy and the negative of the similarity carries more informative signal, introducing a single parameter (described in the appendix section A.2.4) enables the model to account for these more complex conditions.

DGAT combines geometric decomposition with neural attention and gating, mirroring the structure of an Iwasawa-like decomposition that provides strong theoretical grounding and interpretability. Beyond the empirical results presented here, this formulation may serve as a basis for future extensions and adaptations.

# 6 CONCLUSION AND FUTURE WORK

We introduced DGAT, a novel Graph Neural Network architecture that augments traditional similarity-based attention with a dissimilarity-aware mechanism grounded in orthogonal projection and geometric reasoning (Shen et al. (2025)). By explicitly incorporating angular contrast and orthogonal components, DGAT offers a more expressive framework for modeling node interactions, particularly in settings where similarity alone is insufficient.

Empirical evaluations across benchmark datasets demonstrated that DGAT consistently improves performance in node classification, graph classification, and link prediction tasks. The dissimilarity attention not only enhances representational capacity but also mitigates oversmoothing and improves robustness in heterogeneous or symmetric structures (Xu et al. (2022)).

Rather than claiming universal superiority, we position DGAT as a principled framework that addresses fundamental limitations in how graph attention models relational geometry. DGAT should be viewed as complementary with other approaches. Since it operates purely at the embedding level,

DGAT can be integrated with frequency-based methods, contrastive approaches, or newer attention variants, improving their performance while providing geometric interpretability.

Looking ahead, several research directions emerge. First, we plan to explore adaptive gating mechanisms that dynamically modulate the contribution of similarity and dissimilarity during training. This could further improve interpretability and task-specific tuning. Second, DGAT's geometric foundation makes it amenable to extension into other geometric settings, such as hyperbolic manifolds (Cui & Sonthalia (2022)) or non-Archimedean metric spaces. In such spaces, the decomposition structure of Iwasawa or analogous factorizations may allow us to define new forms of attention over non-Euclidean or symbolic domains.

We observe that while DGAT performs as intended at shallow depths, stacking multiple layers amplifies the dissimilarity noise, degrading performance. In future work, we plan to explore denoising strategies and stability improvements for deeper DGAT architectures.

We also foresee applications in domains beyond classical GNN benchmarks. Examples include reasoning over temporal or directed graphs, modeling causal structure with antisymmetric attention kernels, introducing chaotic behavior, and extending the wedge-based dissimilarity to relational data. We posit that DGAT can effectively model chaotic, node-dependent, and anti-natural interactions (irregular, non-transitive, or discontinuous relationships) through node-specific parameters such as unique embeddings within the gating mechanism due to its ability to dinamically separate the two attention components, although empirical validation of these scenarios is left for future work. Overall, DGAT represents not only an architectural innovation, but also a foundation for more principled, geometry-aware, and interpretable message passing networks.

## 7 LLM Usage

A large language model (LLM) was used solely to assist with grammar, writing, and checking the clarity of ideas, including suggestions related to mathematical concepts. All scientific contributions, including the conception, design, implementation, and evaluation of the DGAT model, as well as the interpretation of results, are original and were independently produced by the authors.

## 8 Reproducibility Statement

The code is fully available in the link found in the appendix section A.5. To reproduce the results, check the different tests present in the repository and follow the steps described in the README.md inside the repository.

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

# A APPENDIX

## A.1 HYPERPARAMETERS AND MODEL DETAILS

### A.1.1 HYPERPARAMETERS

Table 3: Hyperparameters used for GCN across homophilic datasets.

| Hyperparameter | MolHIV | Proteins | PPA | DDI |
|---|---|---|---|---|
| Learning rate | 0.01 | 0.005 | 0.01 | 0.005 |
| Dropout | 0.3 | 0.3 | 0.1 | 0.2 |
| Weight decay | 0 | 0 | 0 | 0 |
| Hidden dim | 256 | 256 | 128 | 64 |
| Step Size | 50 | 50 | 50 | 0 |
| Gamma | 0.5 | 0.5 | 0.5 | 0 |

Table 4: Hyperparameters used for GraphSAGE across homophilic datasets.

| Hyperparameter | MolHIV | Proteins | PPA | DDI |
|---|---|---|---|---|
| Learning rate | 0.01 | 0.005 | 0.01 | 0.005 |
| Dropout | 0.3 | 0.3 | 0.1 | 0.2 |
| Weight decay | 0 | 0 | 0 | 0 |
| Hidden dim | 256 | 256 | 128 | 64 |
| Step Size | 50 | 50 | 50 | 0 |
| Gamma | 0.5 | 0.5 | 0.5 | 0 |

Table 5: Hyperparameters used for FAGCN across homophilic datasets.

| Hyperparameter | MolHIV | Proteins | PPA | DDI |
|---|---|---|---|---|
| Learning rate | 0.01 | 0.005 | 0.01 | 0.005 |
| Dropout | 0.3 | 0.3 | 0.1 | 0.2 |
| Weight decay | 0 | 0 | 0 | 0 |
| Hidden dim | 256 | 256 | 128 | 64 |
| Step Size | 50 | 50 | 50 | 0 |
| Gamma | 0.5 | 0.5 | 0.5 | 0 |

Table 6: Hyperparameters used for GAT across homophilic datasets.

| Hyperparameter | MolHIV | Proteins | PPA | DDI |
|---|---|---|---|---|
| Learning rate | 0.005 | 0.01 | 0.01 | 0.005 |
| Dropout | 0.3 | 0.3 | 0.3 | 0.5 |
| Weight decay | 0 | 0 | 0 | 0 |
| Num heads | 8 | 1 | 8 | 2 |
| Hidden dim | 256 | 256 | 128 | 64 |
| Step Size | 50 | 50 | 50 | 0 |
| Gamma | 0.5 | 0.5 | 0.5 | 0 |

Table 7: Hyperparameters used for GATv2 across homophilic datasets.

| Hyperparameter | MolHIV | Proteins | PPA | DDI |
|---|---|---|---|---|
| Learning rate | 0.005 | 0.01 | 0.01 | 0.005 |
| Dropout | 0.3 | 0.3 | 0.3 | 0.5 |
| Weight decay | 0 | 0 | 0 | 0 |
| Num heads | 8 | 1 | 8 | 2 |
| Hidden dim | 256 | 256 | 128 | 64 |
| Step Size | 50 | 50 | 50 | 0 |
| Gamma | 0.5 | 0.5 | 0.5 | 0 |

Table 8: Hyperparameters used for Polynomer across homophilic datasets.

| Hyperparameter | MolHIV | Proteins | PPA | DDI |
|---|---|---|---|---|
| Learning rate | 0.01 | 0.005 | 0.01 | 0.005 |
| Dropout | 0.3 | 0.3 | 0.1 | 0.2 |
| Weight decay | 1e-5 | 0 | 0 | 0 |
| Num heads | 2 | 2 | 1 | 1 |
| Hidden dim | 256 | 256 | 128 | 64 |
| global heads | 2 | 2 | 2 | 2 |
| Step Size | 50 | 50 | 50 | 0 |
| Gamma | 0.5 | 0.5 | 0.5 | 0 |

Table 9: Hyperparameters used for Graph Transformer across homophilic datasets.

| Hyperparameter | MolHIV | Proteins | PPA | DDI |
|---|---|---|---|---|
| Learning rate | 0.005 | 0.01 | 0.01 | 0.005 |
| Dropout | 0.3 | 0.3 | 0.3 | 0.5 |
| Weight decay | 0 | 1e-8 | 0 | 0 |
| Num heads | 8 | 1 | 8 | 2 |
| Hidden dim | 256 | 256 | 128 | 64 |
| Step Size | 50 | 50 | 50 | 0 |
| Gamma | 0.5 | 0.5 | 0.5 | 0 |

Table 10: Hyperparameters used for Exphormer across homophilic datasets.

| Hyperparameter | MolHIV | Proteins | PPA | DDI |
|---|---|---|---|---|
| Learning rate | 0.01 | 0.005 | 0.01 | 0.005 |
| Dropout | 0.3 | 0.3 | 0.1 | 0.2 |
| Weight decay | 1e-5 | 0 | 0 | 0 |
| Num heads | 2 | 2 | 1 | 1 |
| Hidden dim | 256 | 256 | 128 | 64 |
| Step Size | 50 | 50 | 50 | 0 |
| Gamma | 0.5 | 0.5 | 0.5 | 0 |

Table 11: Hyperparameters used for GARN across homophilic datasets. The temperature was made trainable

| Hyperparameter | MolHIV | Proteins | PPA | DDI |
|---|---|---|---|---|
| Learning rate | 0.01 | 0.005 | 0.01 | 0.005 |
| Dropout | 0.3 | 0.3 | 0.1 | 0.2 |
| Weight decay | 0 | 0 | 0 | 0 |
| Num heads | 2 | 2 | 1 | 2 |
| Hidden dim | 256 | 256 | 128 | 64 |
| Step Size | 50 | 50 | 50 | 0 |
| Gamma | 0.5 | 0.5 | 0.5 | 0 |
| $\eta$ | 0.8 | 0.8 | 0.0 | 0.8 |

Table 12: Hyperparameters used for DGAT in all versions (DGAT-Transformer, DGAT-Transformer-Gram, and DGATv2) across homophilic datasets. DGATv2 requires 2 heads in DDI to reproduce the results. For Gate-W-orth in DGAT-Transformer we have the option to use a learned vector instead of scalar.

| Hyperparameter | MolHIV | Proteins | PPA | DDI |
|---|---|---|---|---|
| Learning rate | 0.01 | 0.005 | 0.01 | 0.005 |
| Dropout | 0.3 | 0.3 | 0.1 | 0.2 |
| Weight decay | 0 | 0 | 0 | 0 |
| Num heads | 2 | 1 | 1 | 1 |
| Hidden dim | 256 | 256 | 128 | 64 |
| Step Size | 50 | 50 | 50 | 0 |
| Gamma | 0.5 | 0.5 | 0.5 | 0 |
| Inner-orth | No | No | Yes | Yes |
| Gate-W-orth | Yes | Yes | Yes | No |
| Gate-torsion | No | No | No | No |

Table 13: Hyperparameters used for all baselines in the heterophilic datasets. The same hyper-parameters are used for our models, with the exception of the number of heads, that varies along models and benchmarks. In Minesweeper was set to 6 for GAT, 3 for GATv2, 4 for GTransf, and 3 for DGAT in its three variants. For the rest of the datasets, DGAT variants required half the number of heads used by the baselines.

| Hyperparameter | Minesweeper | Roman-Empire | Amazon-Ratings | Questions |
|---|---|---|---|---|
| Learning rate | 0.005 | 0.005 | 0.001 | 0.001 |
| Dropout | 0.2 | 0.3 | 0.3 | 0.3 |
| Weight decay | 0 | 0 | 0 | 0 |
| Layers | 6 | 10 | 2 | 3 |
| Num heads | 4 | 4 | 2 | 2 |
| Hidden dim | 64 | 256 | 512 | 512 |
| Inner-orth | No | No | Yes | Yes |
| Gate-W-orth | Yes | Yes | Yes | Yes |
| Gate-torsion | No | No | No | No |
| Global heads | 2 | 2 | 2 | 2 |

### A.1.2 DGAT INTUITION

Consider a toy graph, represented in Figure 3, where nodes AA, BB, and CC all have nearly identical features due to their common neighbors. A Similarity based model (i.e GAT) would assign nearly uniform attention weights, treating them as indistinguishable. However, with DGAT, the dissimilarity component captures that BB differs from CC in its angular relationship to AA, enabling richer representation of roles or anomalies.

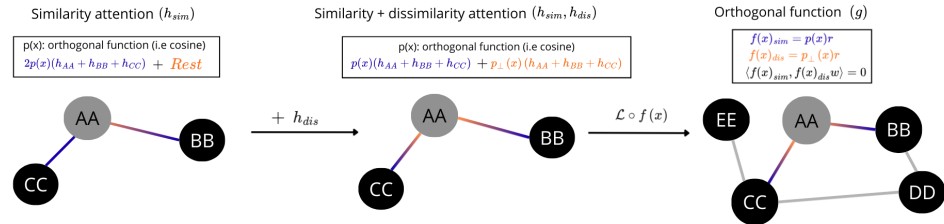

Figure 3: Progression of DGAT attention with similarity and dissimilarity components. Similarity-based attention alone often ignores information from orthogonal components. To mitigate this, it is paired with a complementary dissimilarity-based attention that counterbalances excessive similarity. Optionally, a global orthogonality constraint can then be applied across the entire graph, enforcing functional orthogonality between similarity and dissimilarity components.

Also another issue will arise when the attention of two nodes AA and BB is compared with the attention between AA and CC. The node BB has a higher norm but less cosine similarity with a normalized AA, and even so it will "collide" with the combination of AA and CC even though they have more cosine similarity because the norm and angle are not properly handled. This generates issues in the computation of the attentions (Lee et al. (2023b)) easily fixed with the dissimilarity component. In addition, in Figure 3 we can also use the whole graph (DD and EE are part of the graph but not neighbors of AA) as a space where the similarity and dissimilarity functions are orthogonal, giving a richer representation of the embeddings.

### A.1.3 ARCHITECTURE

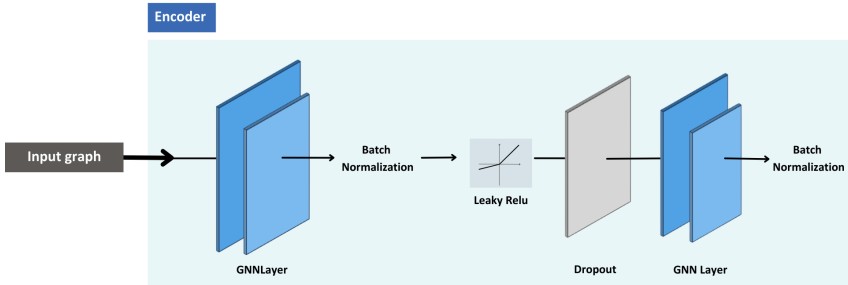

Figure 4: Common architecture for GAT, GATv2, GraphTransfomer and DGAT in all versions (DGAT-Transformer, DGAT-Transformer-Gram, and DGATv2)

### A.2 MATHEMATICAL BACKGROUND FOR DGAT

### A.2.1 GEOMETRICAL INTERPRETATION OF DGAT

Lemma 1: Let $\mathcal{M} \subset \mathbb{R}^d$ be a smooth manifold of node embeddings. For each node $i \in \mathcal{V}$, consider its neighborhood $\mathcal{T}_i \subset \mathcal{V}$ and the associated fibre bundle

$$\mathcal{R}_i = \bigcup_{j \in \mathcal{T}_i} (V_i \otimes V_j),$$

where $V_i, V_j$ are the feature spaces of nodes $i, j$, respectively.

The mapping $\phi : \mathcal{R}_i \to \mathcal{M}$ from the local fibre bundle to the global manifold must be constructed via orthogonal projections that preserve the geometric structure. Specifically, for each edge $(i, j)$, the mapping is given by:

$$\phi(v_i \otimes v_j) = g_i \cdot (I - P_j)v_i + (1 - g_i) \cdot P_j v_i,$$

where:

$P_j : V_i \to V_j$ is the orthogonal projection operator: $P_j(v_i) = \frac{\langle v_i, v_j \rangle}{\|v_j\|^2} v_j$, $g_i \in [0, 1]$ is a learnable gate parameter, $\cdot (I - P_j)$ projects onto the orthogonal complement.

This mapping satisfies:

1 Injectivity: If $\phi(v_i \otimes v_j) = \phi(v'_i \otimes v'_j)$, then $v_i \otimes v_j = v'_i \otimes v'_j$.

2. Orthogonality: $\langle P_j v_i, (I - P_j)v_i \rangle = 0$.

3. Non-expansiveness: $\|\phi(v_i \otimes v_j) - \phi(v'_i \otimes v'_j)\| \leq \|v_i \otimes v_j - v'_i \otimes v'_j\|$.

Proof:

For orthogonality: $\langle P_j v_i, (I - P_j)v_i \rangle = \langle P_j v_i, v_i \rangle - \langle P_j v_i, P_j v_i \rangle = \|P_j v_i\|^2 - \|P_j v_i\|^2 = 0$.

For non-expansiveness:

$$\|\phi(v_i \otimes v_j) - \phi(v'_i \otimes v'_j)\| = \|g_i[(I - P_j)(v_i - v'_i)] + (1 - g_i)[P_j(v_i - v'_i)]\|$$

$$\leq |g_i| \|(I - P_j)(v_i - v'_i)\| + |1 - g_i| \|P_j(v_i - v'_i)\|$$

$$\leq |g_i| \|v_i - v'_i\| + |1 - g_i| \|v_i - v'_i\| = \|v_i - v'_i\| \leq \|v_i \otimes v_j - v'_i \otimes v'_j\|$$

since $\|P_j\| = 1$ and $\|I - P_j\| = 1$ for orthogonal projections, and $g_i \in [0, 1]$ .

For injectivity, assume

$$\phi(v_i \otimes v_j) = \phi(v'_i \otimes v'_j)$$

Then both the parallel and orthogonal components must match separately, implying $v_i = v_i'$ and the relationship to $v_j$ is preserved.

Corollary 1.1: The gate parameter $g_i$ learns the optimal connection between local neighborhood geometry and global manifold structure, adapting to the curvature of the embedding space.

Regarding convergence, if the model is trained correctly, the gating parameter ($\lambda$) can augment, reduce, or suppress dissimilarity propagation, effectively reverting the model to a standard GATv2 or Graph Transformer when needed. This ensures that, in the worst case, DGAT can self-stabilize during training and possesses mechanisms to control divergence.

This geometric foundation increases the expressivity through destination-node regulation instead of focusing only on the neighbors' embedding. Unlike methods that filter information at the source, DGAT ensures dissimilar information contributes meaningfully to the target embedding, leading to more stable propagation dynamics. We also make use of our adaptive scaling to control these dynamics. By allowing unnormalized, positive-valued components, DGAT adapts to neighborhood complexity rather than forcing artificial normalization. When the adaptation includes orthogonal information, we can also provide noise robustness. The manifolds perspective naturally handles unreliable graphs where simple similarity measures break down.

### A.2.2 Dissimilarity Attention via Orthogonal Projection and Iwasawa Decomposition

DGAT introduces a novel attention mechanism that explicitly incorporates dissimilarity, complementing the classical similarity-based approaches used in prior attention mechanisms (Buterez et al. (2025)) such as GAT. Although traditional dot product-based attention captures feature alignment, it often fails to account for contrasting or orthogonal differences between node representations. This limits expressivity, particularly in graphs where critical interactions arise from dissimilar neighbors rather than similar neighbors.

To address this, DGAT leverages orthogonal projection to quantify angular contrast. By computing the component of one node's feature vector that is orthogonal to another, DGAT captures a dissimilarity signal that is overlooked by standard dot products. This enhances the model's ability to distinguish structurally or functionally distinct nodes, particularly in heterophilic graphs.

Interestingly, the formulation of DGAT resonates with the mathematical structure of the **Iwasawa decomposition** (Lenz et al. (2010)), a well-known factorization in Lie group theory. The Iwasawa decomposition expresses a matrix as a product of three distinct components:

$$G = K \cdot A \cdot N \tag{9}$$

Where:

- K: An orthogonal matrix representing rotations. In DGAT, this corresponds to the dual attention components (similarity and dissimilarity), which encode both feature alignment and angular contrast in the embedding space.
- A: A diagonal matrix capturing scaling or dispersion. In DGAT, this role is realized through the explicit difference vector , which amplifies the influence of dissimilar components in message passing.
- N: A unipotent upper-triangular matrix introducing shear or shift. In DGAT, this is modeled via the learnable gating mechanism that adaptively weights the contribution of similarity and dissimilarity based on the graph structure and edge attributes.

This decomposition provides a principled and interpretable framework for disentangling different aspects of relational interactions in graphs. DGAT mirrors this structure by embedding similarity (K), difference (A), and adaptive control (N) as modular and composable components of its architecture. Table 14 outlines these components and their correspondence to the elements of the Iwasawa decomposition:

Moreover, although DGAT is implemented in Euclidean (Archimedean) space, the decomposition it follows can be generalized. Replacing the dot product and orthogonal projection (Chen et al.

Table 14: Components of the Iwasawa decomposition and their and their analogues in DGAT.

| Component | Interpretation in DGAT |
|---|---|
| $K$ | Dual attention (similarity and dissimilarity) |
| $A$ | Dimension-wise controlled difference vector $h_i - h_j$ or $h_i^{\perp}$ (w-orthogonal dispersion) |
| $N$ | Gating mechanism $g_{ij}$ (adaptive shift) |

(2024)) with other similarity/dissimilarity functions would allow DGAT to extend naturally into non-Archimedean geometries—such as hyperbolic or ultrametric spaces—or even symbolic domains. This positions DGAT as a highly general and theoretically grounded framework for message passing in diverse graph structures.

Through this geometric lens, DGAT is not just an architectural modification but an instance of a broader class of attention models structured by algebraic decomposition. It offers both empirical utility and a foundation for further theoretical exploration of attention in geometric and group-theoretic terms (Matveev et al. (2020)).

### A.2.3 ORTHOGONAL ENERGY AND DIFFERENTIAL STRUCTURE IN DGAT

We reformulate the DGAT message passing to make explicit the orthogonality and geometric balance (Chen et al. (2024)) between the similarity and dissimilarity components, ensuring energy preservation and elliptical propagation. Note that computing the maximum corresponds to treating the dimensions as common element while keeping the differences in another dimensions.

Starting from the DGAT update rule (* denotes elementwise product):

$$\sum \alpha_{ij}^{sim} * h_j^l + g(\sum a_{ij}^{dis} * h_i^l)$$

we define the similarity direction $u = h_j^l$ and the dissimilarity direction $v = h_i^l$. Due to the max operator, combined with the difference $\alpha_{dis} - \alpha_{sim}$, naturally generates two effective directions:

$$\langle \alpha_{sim} * u, \ \alpha_{dis} * v \rangle = \alpha_{sim} * \alpha_{dis} * \langle h_j^l + h_i^l, h_i^l - h_j^l \rangle$$

This dot product, under angular decomposition (note that $h_i$ and $h_j$ due to attention and gate transformations are orthogonal), yields:

$$\propto \frac{\sin(2\theta)}{2} \left( \|h_i^l\|^2 - \|h_j^l\|^2 \right)$$

which peaks at $\theta = \frac{\pi}{4}$, indicating optimal energy spread when similarity and dissimilarity are balanced.

The induced area, modeled by the norm of the cross product, is:

$$\|\alpha_{sim} * u \times \alpha_{dis} * v\| = \frac{\sin(2\theta)}{2} \|h_i^l \times h_j^l\|$$

again maximal at angular equilibrium. This constructs a triangle in feature space, bounded by $h_i$, $h_j$, and their averaged and differential combinations.

This structure naturally aligns with the Cayley–Menger determinant (de Salles Neto et al. (2021)), used for computing areas and volumes from pairwise distances:

$$A_\triangle = \sqrt{\frac{1}{4}d_{ij}^2 d_{jk}^2 - \left( \frac{d_{ik}^2 + d_{ij}^2 - d_{jk}^2}{2} \right)^2}$$

DGAT's update lies within such a simplex, defined by embedding distances and their angular contrasts.

**Connection to Curl and Discrete Rotation**    DGAT's dissimilarity term also resembles the integral form of curl from vector calculus:

$$\|\nabla \times \mathbf{F}(p)\| \approx \lim_{A \to 0} \frac{1}{|A|} \oint_{\partial A} \mathbf{F} \cdot d\mathbf{l} \approx \frac{1}{|\partial A|} \sum_{\partial A} \|\mathbf{F}\| \|d\mathbf{l}\| \sin(\theta)$$

which captures the circulation of a vector field around a local region. This formulation interprets curl as angular deviation integrated along a path, scaled by the perimeter of a neighborhood.

The differential definition of curl,

$$\|\nabla \times \mathbf{F}\| = \begin{vmatrix} i & j & k \\ \frac{d}{dx} & \frac{d}{dy} & \frac{d}{dz} \\ F_x & F_y & F_z \end{vmatrix} = \left\| \frac{d\mathbf{F}}{ds} \right\| \sin(\theta)$$

relies on coordinate-specific cross products and partial derivatives, and is strictly defined only in 3D and 7D spaces due to algebraic constraints on vector products.

In contrast, DGAT's dissimilarity update:

$$softmax(\|h_i \wedge h_j\|) \approx softmax(\|h_i - h_j\| sin(\theta))$$

$$softmax(\|h_i - h_j\| sin(\theta)) = \|dh\| sin(\theta)$$

$$\alpha_{ij}^{dis}\|h_i\| \approx \|h_i\|\|dh\| \sin(\theta)$$

It can be noted that is structurally closer to the norm of the integral curl, not the differential version. It aggregates angular deviation over local edges in a graph neighborhood, analogous to the discretized line integral of a vector field along a loop.

This similarity justifies interpreting DGAT as a discrete, norm-preserving, rotation-aware operator. It approximates geometric flow across irregular graphs without requiring continuous manifolds or differentiable structure, making it extensible to high-dimensional or non-Euclidean domains.

This formulation, through its parameterization, also mimics the structure of eigenfunctions. Since sine and cosine form a canonical pair of orthogonal basis functions, DGAT's similarity–dissimilarity decomposition inherits the capacity to approximate orthogonal function systems. In this sense, DGAT can be seen not only as a discrete, rotation-aware operator, but also as a mechanism for simulating orthogonal expansions of signals and vector fields. This dual perspective reinforces its suitability for modeling geometric flows in graphs while extending naturally to high-dimensional or non-Euclidean domains.

A further connection arises in the case of rigid graphs, where the proposed orthogonality can represent generator rules more effectively than purely similarity-based strategies. In particular, Cayley (Janssen & MacKeigan (2020)) and fractal graph structures can be naturally modeled through cyclic patterns that depend on geometric constraints, applied locally across the leaves of the graph.

**Connection to Sturm-Liouville and bassel problems** The w-orth gate in DGAT relates naturally to Sturm–Liouville theory, since it relies on families of orthogonal functions to regulate and separate the different message-passing branches. Interpreting the neighborhood structure of node embeddings as a discrete manifold, the orthogonal component can be viewed as selecting eigenfunctions of a local Sturm–Liouville operator defined over this manifold. These eigenfunctions provide an orthogonal basis that reflects the intrinsic geometric structure encoded in the embeddings.

This viewpoint also connects DGAT to polynomially generated Lie algebras: the eigenfunctions of Sturm–Liouville operators form orthogonal bases that are closed under specific algebraic operations (problem settings). This allows DGAT to incorporate discrete analogues of Lie algebraic transformations directly into its attention mechanisms.

**Connection to signal processing** During message passing, a key challenge is preserving meaningful differences between nodes. In many existing methods (e.g., GARN , FAGCN, ACMGNN), dissimilarity information is often suppressed or removed. In contrast, DGAT explicitly aims to capture and diffuse non-common information without losing the shared information between nodes. This removes the "zero-sum" behavior, allowing controlled diffusion of information that lies between nodes' probabilistic gates. The gating mechanism controls the diffusion, ensuring it does not grow excessively while maintaining orthogonal contributions. This is the main source of DGAT's

non-expansive behavior. The orthogonal projection ensures that the diffused information adds independent contributions to embeddings, preventing unbounded growth. The torsion gate removes combinations that are not suited for the problem at hand, which can be beneficial or neutral depending on the dataset. And last the dropout acts as a crucial stabilizer, connecting these components and helping ensure that learned information is robust.

DGAT introduces a fundamentally different geometric perspective: instead of treating neighborhoods as bags of independent signals, we model them as local manifolds where both parallel (similar) and orthogonal (dissimilar) components carry meaningful relational information. Our unified dual-attention mechanism preserves these geometric correlations by operating directly on embeddings rather than decomposing them into frequency bands.

### A.2.4 EFFICIENT IMPLEMENTATION AND NORMALIZATION

The dissimilarity term is computed without requiring explicit pairwise vector differences. Instead, it relies on dot products and precomputed norms, allowing for batched and GPU-efficient implementation. As a practical and numerically stable approximation to orthogonal projection, we use the squared sine formulation:

$$\sin^2(\theta) = 1 - \cos^2(\theta)$$

which avoids the need for computing projection residuals directly, while retaining angular sensitivity.

Notably, cosine similarity exhibits flat gradients when vectors are nearly aligned or orthogonal (He & Cheng (2025)). In these regions, the derivative of the cosine function approaches zero, which limits the ability of attention to distinguish between similar or ambiguously positioned vectors. By incorporating the $\sin^2(\theta)$ formulation, we obtain sharper gradients near the boundaries ($\theta \approx 0$ or $\theta \approx \pi/2$), improving sensitivity to angular contrast and resulting in more robust dissimilarity-based attention.

To further simplify computation and reduce intermediate tensor overhead, we approximate wedge and cross products using orthogonal projection via Gram decomposition:

$$u_\perp = v - \frac{\langle v, u \rangle}{\langle u, u \rangle + \epsilon} u \tag{10}$$

This operation isolates the component of $v$ that is orthogonal to $u$, and can be used to compute directional dissimilarity without explicitly forming a cross product or wedge product tensor.

We note that both wedge and cross product dissimilarities reduce to a form proportional to $\|b\| \sin(\theta)$, as shown below:

$$\frac{\|a \wedge b\|}{\|a\|} \simeq \frac{\|a \times b\|}{\|a\|} = \|b\| \sin(\theta) \tag{11}$$

which simplifies to an efficient projection term:

$$\left\| a - \frac{\langle a, b \rangle}{\langle b, b \rangle} b \right\|$$

In Equation 10, we observe that for a fixed center node $a$, the norm $\|a\|$ remains constant across its neighbors, allowing us to factor it out without loss of generality, further simplifying computation.

The final attention formulation combines similarity and dissimilarity components with minimal overhead while maintaining geometric expressiveness. This is done in combination with the gates (Equation 4). The first gate, w-orth, provides a mechanism to model Sturm–Liouville-based functional orthogonality Vigouroux et al. (2025)). Through this parameter, the model can adapt to problems where embeddings are characterized by manifold-like curves. The second gate, used in conjunction with the first, captures the nonlinear torsion or Lie–bracket structure of the problem (The parameter makes it adaptive). Together, these two gates enable the model to handle problems with rigid embeddings that nevertheless require additional flexibility, without increasing the signal-to-noise ratio or losing the structural information encoded in the embedding relationships.

The w-orth constraint can both help and hurt, depending on the data and training dynamics. Empirical results indicate that it is particularly beneficial in scenarios where nodes require strong feature

separation or where discriminative signals are inherently weak. In such cases, the gate's effect is complemented by the model's internal orthogonalization dynamics, which help maintain robustness even under sparsity or local heterogeneity.

In case of fully noisy enviroments the addition of a scaler parameter in the computation of the $\alpha_{ij}^{dis}$ would be able to model more complex scenarios.

$$\beta = \sum_{p=1}^{dim(x_i)} \boldsymbol{W} * (x_{i,m_p}^{\perp} - x_{i,mean\mathcal{N}(x_i)_p}^{\perp})$$

This parameter allows to model noise and reduce it or amplify it on demand. $x_i$ refers to the orthogonal of the link $m$ in the neighborhood $\mathcal{N}(x_i)$ where $x_i^{\perp}$ represents the orthogonal of $x_i^{\perp}$ in that specific link. $x_{i,mean\mathcal{N}(x_i)}^{\perp}$ is the mean orthogonal component across all edges incident to $x_i$

The new formula requires changing the softmax in equations 6 and 8 to the following formulation:

$$\alpha_{ij}^{dis} = \begin{cases} dropout(\text{softmax}\,(\lambda_{dis} * \beta * \boldsymbol{W} * \alpha_{dis}^{gram})) & if\ training \\ \text{softmax}\,(\lambda_{dis} * \beta * \boldsymbol{W} * \alpha_{dis}^{gram}) & otherwise \end{cases} \qquad (12)$$

The key distinction lies in how we handle dissimilarity: where prior work typically treats it as noise to be removed or suppressed, DGAT models it as a geometric integral over the local manifold. This allows us to identify and integrate differences in a controlled manner through functional orthogonal and torsion gates, without enforcing artificial normalization constraints that force similarity and dissimilarity to sum to one.

### A.3 ABLATION STUDY

The Table 15 reports the performance of DGAT variants (DGATv2, DGAT-Transformer, and DGAT-Transformer-Gram) under different gate configurations and with or without the dissimilarity component. The ablation study was conducted on heterophilic graphs, as performance on homophilic graphs is more stable; the true benefits of the dual-attention and gating mechanisms are most apparent in heterophilic benchmarks. The combinations of each gate with dissimilarity highlight the necessity of each configuration depending on the input dataset. The entries in 15 list the gates and elements used in parentheses, while those without parentheses correspond to the basic models.

Table 15: Results table of ablated models. 1: inner orth, 2: w_orth gate, 3: torsion gate

| gates ablation | Minesweeper | Roman-Empire | Amazon-Ratings | Questions |
|---|---|---|---|---|
| GATv2 (baseline) | 0.9140 ± 0.0082 | 0.8980 ± 0.0032 | 0.5520 ± 0.0067 | 0.9723 ± 0.0009 |
| DGATv2 | 0.8980 ± 0.0020 | 0.8940 ± 0.0031 | 0.5520 ±0.0030 | 0.9724 ± 0.0009 |
| DGATv2 (1) | 0.9050 ±0.0032 | 0.8980 ± 0.0037 | 0.5514 ±0.0042 | 0.9724 ± 0.0009 |
| DGATv2 (1,2) | 0.9145 ±0.0052 | 0.9030 ± 0.0043 | **0.5581** ±0.0064 | **0.9728** ± 0.0009 |
| DGATv2 (1,2,3) | 0.9141± 0.0093 | 0.9020 ± 0.0021 | 0.5540 ±0.0062 | 0.9723 ± 0.0009 |
| DGATv2 (2) | **0.9148** ± 0.0089 | **0.9174** ± 0.0040 | 0.5523 ±0.0052 | 0.9726 ± 0.0009 |
| DGATv2 (2,3) | 0.9127 ± 0.0071 | 0.9140 ± 0.0029 | 0.5577 ± 0.0044 | 0.9721 ± 0.0009 |
| GraphTransformer (baseline) | 0.8987 ± 0.0052 | 0. 8051 ± 0.0034 | 0.5590 ± 0.0052 | 0.9712 ± 0.0009 |
| DGAT-Transf | 0.9021 ± 0.0020 | 0.8530 ± 0.0031 | 0.5552 ±0.0081 | 0.9718 ± 0.0009 |
| DGAT-Transf (1) | 0.8722 ± 0.0020 | 0.8069 ± 0.0031 | 0.5180 ±0.0081 | 0.9691 ± 0.0009 |
| DGAT-Transf (1,2) | **0.9092** ± 0.0052 | **0.9056** ± 0.0043 | **0.5582** ±0.0055 | **0.9723** ± 0.0009 |
| DGAT-Transf (1,2,3) | 0.9080± 0.0093 | 0.8512 ± 0.0021 | 0.5561 ±0.0071 | 0.9720 ± 0.0009 |
| DGAT-Transf (2) | 0.8958 ± 0.0089 | 0.9052 ± 0.0089 | 0.5551 ±0.0042 | 0.9719 ± 0.0007 |
| DGAT-Transf (2,3) | 0.9038 ± 0.0071 | 0.8540 ± 0.0029 | 0.5538 ± 0.0044 | 0.9721 ± 0.0009 |
| DGAT-Transf-Gram | 0.9038 ± 0.0032 | 0.8570 ± 0.0037 | 0.5547 ±0.0046 | 0.9722 ± 0.0009 |
| DGAT-Transf-Gram (1) | 0.8214 0.0032 | 0.7832 ± 0.0037 | 0.5240 ±0.0046 | 0.9715 ± 0.0009 |
| DGAT-Transf-Gram (1,2) | **0.9095** ± 0.0089 | **0.9010** ± 0.0043 | **0.5580** ±0.0064 | **0.9725** ± 0.0009 |
| DGAT-Transf-Gram (1,2,3) | 0.9043± 0.0093 | 0.8563 ± 0.0021 | 0.5573 ±0.0062 | 0.9722 ± 0.0009 |
| DGAT-Transf-Gram (2) | 0.9045 ± 0.0052 | 0.9018 ± 0.0040 | 0.5553 ±0.0052 | 0.9721 ± 0.0009 |
| DGAT-Transf-Gram (2,3) | 0.9082 ± 0.0071 | 0.8684 ± 0.0029 | 0.5557 ± 0.0060 | 0.9724 ± 0.0009 |

## A.4 EFFICIENCY OF THE MODEL

We have compared the execution times of our models with all the baselines on the eight datasets considered. A summary of this comparison is shown in Table 16.

All DGAT variants (DGATv2, DGAT-transf and DGAT-transf-gram) are somewhat less efficient, to varying degrees, than the Linear type attention Transformer models, with the exception of Node-former, which exhibits execution times similar to —and sometimes even higher than— those of the DGAT variants.

However, several points should be highlighted. First, in the comparison with Graph Transformer, the percentage increase in execution time is very small for the datasets with higher computational cost, and the larger percentage increases occur mainly in the lower-cost datasets, where they are less impactful. Second, in the comparison with GATv2, the difference in execution time is larger, although DGATv2 consistently outperforms GATv2 in all datasets (as shown in Table 2 of the paper). Third, we would also like to point out that DGAT variants are not yet optimized, and future work could further improve its efficiency. Therefore, it can be argued that the performance gain can outweigh the additional latency, given that no memory bottlenecks were observed.

Table 16: Execution time for each model. The values are expressed in seconds per epoch.

| Time/epoch (sec/epoch) | MolHIV | Proteins | PPA | DDI |
|---|---|---|---|---|
| GCN | 180 | 9 | 20 | 10 |
| GraphSage | 200 | 10 | 20 | 10 |
| FAGCN | 180 | 9 | 20 | 10 |
| GAT | 205 | 10 | 22 | 10 |
| GATv2 | 211 | 10 | 25 | 12 |
| GraphTransformer | 220 | 11 | 25 | 14 |
| Exphormer | 220 | 11 | 26 | 14 |
| Polynormer | 209 | 11 | 23 | 13 |
| GARN | 220 | 13 | 27 | 14 |
| DGATv2 | 222 | 13 | **27** | 14 |
| DGAT-Transformer | 221 | 15 | **30** | 15 |
| DGAT-Transformer-Gram | 221 | 13 | **27** | 14 |
| Nodeformer* | 221 | 15 | **29** | 14 |

| Time/epoch (sec/epoch) | Minesweeper | Roman-Empire | Amazon-Ratings | Questions |
|---|---|---|---|---|
| GCN | 0.3 | 10 | 10 | 9 |
| GraphSage | 0.3 | 12 | 12 | 12 |
| FAGCN | 0.3 | 10 | 10 | 9 |
| GAT | 0.5 | 10 | 10 | 12 |
| GATv2 | 0.9 | 12 | 14 | 14 |
| GraphTransformer | 1 | 15 | 17 | 16 |
| Exphormer | 1 | 17 | 18 | 19 |
| Polynormer | 0.8 | 13 | 14 | 14 |
| GARN | 1.5 | 19 | 20 | 18 |
| DGATv2 | 1.7 | **20** | **20** | **19** |
| DGAT-Transformer | 1.7 | **20** | **20** | **21** |
| DGAT-Transformer-Gram | 1.7 | **19** | **19** | **20** |
| Nodeformer* | 1.7 | **22** | **21** | **21** |

## A.5 CODE AND REPRODUCIBILITY

The code for some experiments is publicly available at the supplementary materials. To reproduce the experiments, please follow the instructions provided in the repository's README. The specific hyperparameters used for each dataset are detailed in Section A.1.1.

