# OpenReview forum: "Difference-Based Graph Attention Networks: A Dual Attention Mechanism for Similarity and Dissimilarity in Graph Learning"
_ICLR.cc/2026/Conference — Submitted to ICLR 2026_

### Official Review · Reviewer_Jp3V · 2025-10-20

**Soundness:** 3
**Presentation:** 3
**Contribution:** 3
**Rating:** 4
**Confidence:** 3

**Summary:**

This paper introduces Difference-Based Graph Attention Network (DGAT), which is a novel GNN architecture that extends traditional similarity-based attention with an additional dissimilarity-aware attention pathway. DGAT models both similarity (via cosine or additive attention) and dissimilarity (via orthogonal projection and wedge-product approximations) within a unified geometric framework. The authors provide a theoretical grounding based on the Iwasawa-Cayley decomposition from Lie group theory, showing that similarity and dissimilarity components correspond to orthogonal, scaling, and shifting operations.

**Strengths:**

This paper introduce a dissimilarity-based attention mechanism into graph attention networks for the first time, addressing the long-standing limitation of relying solely on feature similarity. By modeling both similarity and dissimilarity between nodes, the proposed DGAT significantly enhances representation expressiveness. The paper provides a geometric interpretation by linking DGAT to the Iwasawa-Cayley decomposition, offering strong theoretical grounding and interpretability. Moreover, the DGAT framework is highly flexible and extensible, integrating seamlessly with existing architectures such as GAT, GATv2, and Graph Transformers.

**Weaknesses:**

1. The paper lacks a clear and intuitive diagram or framework illustration that helps readers better understand how the proposed approach works.

2. The paper should clarify how the proposed method differs from or relates to graph contrastive learning, which also uses node similarity and dissimilarity.

3. The motivation is unclear, the paper only mentions that previous models ignored node dissimilarity but does not analyze why dissimilarity was not utilized. Moreover, since the experiments include both homophilic and heterophilic graphs, it is not clear which type the study primarily focuses on. Some baseline models are designed for homophilic graphs, so applying them directly to heterophilic settings may raise questions about the fairness and validity of the comparisons.

4. The baseline methods are outdated and lack comparisons with more recent GNN models such as GARN[1]. In addition, the baselines mainly include attention-based architectures, so it is unclear why other classic methods such as GCN[2], GraphSAGE[3], and JKNet[4] were not included for comparison.

[1] Wang Y, Wen J, Zhang C, et al. Graph aggregating-repelling network: Do not trust all neighbors in heterophilic graphs. Neural Networks, 2024, 178: 106484.

[2] T. N. Kipf and M. Welling. Semi-supervised classification with graph convolutional networks. In 5th International Conference on Learning Representations, 2017.

[3] W. L. Hamilton, Z. Ying, and J. Leskovec. Inductive representation learning on large graphs. In Proceedings of Advances in Neural Information Processing Systems, 2017.

[4] Xu K, Li C, Tian Y, et al. Representation learning on graphs with jumping knowledge networks. In Proceedings of the 35th International Conference on Machine Learning, 2018.

5.The paper lacks an analysis of time and space complexity.

**Questions:**

See weakness.

---

> ### Author Response · Authors · 2025-11-28
>
> We sincerely thank the reviewer for the remarks that pointed out missing aspects of our work.
>
> ---
>
> ### **W1**
>
> We appreciate the reviewer’s comment on this aspect. We have added **Figure 1** to the paper that illustrates how our approach exploits meaningful information that is typically discarded by other models (by using orthogonality in our case). It graphically shows the novelty of this paper compared with previous approaches on this topic.
>
> ---
>
> ### **W2**
>
> We are grateful for this important remark from the reviewer. Although contrastive methods and DGAT share certain conceptual similarities (such as emphasizing complementary signals), they operate under fundamentally different principles and can, in fact, be combined. DGAT is architecturally constrained to enforce geometric coherence through its orthogonality-based design. Contrastive methods, by comparison, are considerably more flexible and may converge to representations that are not geometrically consistent, which can be advantageous for problems involving boundary conditions or non-orthogonal latent structures.
>
> Thus, while **contrastive learning** can approximate some of the effects achieved by DGAT, its behavior is less controlled and may disregard geometric consistency. A particularity that DGAT addresses is avoiding the risk of uncontrolled or “blind” convergence that mixed with known boundary conditions allow DGAT to be complementary with contrastive learning.
>
> This complementarity suggests that DGAT and contrastive methods (as well as the $-\cos\alpha$ family of models) occupy different but compatible roles within the geometric representation space.
>
> ---
>
> ### **W3**
>
> We thank the reviewer for this remark. We acknowledge that the text may not have been sufficiently clear, so we will clarify that the contribution of our model is to leverage dissimilarity rather than simply remove it, extracting useful information from it. The main goal of our study is to address a limitation in attention-based graph models, namely their limited use of node dissimilarity.
>
> While methods such as GARN, FAGCN, ACMGCN, and A2GCN incorporate aspects of dissimilarity, they primarily focus on diminishing or filtering out dissimilar components (typically through terms proportional to $-\cos\alpha$). In contrast, DGAT introduces an **orthogonal component** based on $\sin\alpha$, which is complementary to these existing formulations. This makes DGAT and prior models compatible and combinable: the $-\cos\alpha$ term captures the direction opposite to similarity, while the $\sin\alpha$ term captures orthogonal deviation, enabling a more complete geometric characterization of node interactions.
>
> Our experiments on both homophilic and heterophilic graphs aim to evaluate whether existing graph attention networks and graph Transformers can be adapted to handle both graph types. The results highlight the importance of preserving, rather than discarding, seemingly unnecessary information.
>
> References:
> * [arXiv:2101.00797] Beyond Low-frequency Information in Graph Convolutional Networks
> * [arXiv:2210.07606] Revisiting Heterophily For Graph Neural Networks
> * [DOI:10.1016/j.patcog.2024.110764] A2GCN: Graph Convolutional Networks with Adaptive Frequency and Arbitrary Order
> ---
>
> ### **W4**
>
> We appreciate this thoughtful suggestion and have incorporated additional analysis to address it.
>
> We have included 6 more **baselines**. The results obtained across all benchmarks are shown in Tables 1 and 2 of the paper. The following observations can be made:
>
> - **GCN** and **GraphSAGE** produced results similar to the regular GAT.
> - The challenge with **GARN** is that no public implementation is available. Consequently, we have implemented it ourselves in order to include it as suggested by the reviewer. We are grateful for the suggestion as the good performance of GARN in some scenarios could reinforce the central idea we address in the paper. Even though GARN adopts a different perspective, it overlaps with some of the operations that we are proposing for dissimilarity.
> - We initially focused on older baselines because most modern methods (e.g., Polynormer, Exphormer) rely on GAT or Graph-Transformer-type attention. **Graphormer** introduces novel attention mechanisms but its computational footprint is too demanding for standard GPUs, so we discarded it to focus on lighter attention layers.
>
> References:
> * [arXiv:2303.06147] Exphormer: Sparse Transformers for Graphs
> * [arXiv:2403.01232] Polynormer: Polynomial-Expressive Graph Transformer in Linear Time
> * [arXiv:2106.05234] Do Transformers Really Perform Bad for Graph Representation?
>
> In addition, we have clarified in Table 2 that, for the heterophilic benchmarks, all models were evaluated using JKNet  and ResNet in order to preserve fairness in the comparison and ensure that the models behave as intended.
>
> ---
>
> We greatly appreciate your questions and suggestions, which have helped us continue improving the paper.

---

### Official Review · Reviewer_21ta · 2025-10-26

**Soundness:** 2
**Presentation:** 3
**Contribution:** 2
**Rating:** 2
**Confidence:** 5

**Summary:**

This paper develops a new graph neural network called DGAT. The authors argue that existing attention-based GNNs are built on similarity-based attention mechanism, which is inefficient in capture complex graph information. To this end, the authors develop a new attention module based on orthogonal projections and wedge-prodcut approximations to preserve contrastive relationships between nodes. Experimental results on various datasets show the effectiveness of DGAT on graph data mining tasks.

**Strengths:**

1.This paper is well-organized and easy to follow.

2.The authors provide the theoretical analysis of the proposed method.

3.The proposed DGAT provides new insights for attention-based GRL methods.

**Weaknesses:**

1.The research gap is overclaimed.

2.Mainstream baselines are missing.

3.Some important experiments are missing.

**Questions:**

1.The authors claim the limitation in existing attention-based GRL methods which lack objectivity. There are also several works, such as FAGCN and ACMGNN, which are designed to capture both low-frequency information (similarity) and high-frequency information (dissimilarity) in graph representation learning.

2.In the experiment part, I suggest the authors add more recent graph Transformers as baselines for performance comparison.

3.Moreover, necessary experimental designs such as ablation study and parameter analysis are also required for strengthening the experiment part.

---

> ### Author Response · Authors · 2025-11-24
>
> We thank the reviewer for the detailed analysis of our work. We have followed the suggestions and revised the manuscript accordingly. We appreciate the insightful **questions**, which we have answered below:
>
> ---
>
> **Q1**. We thank the reviewer for highlighting the connection with the cited works. We have added FAGCN and GARN as new baselines (Tables 1 and 2 in the paper). ACMGNN was not included due to the requirement to preprocess the adjacency matrix. Although there is partial methodological overlap, our solution is built on different assumptions. We clarify the distinctions below.
>
> While frequency-based approaches (e.g., FAGCN, ACMGNN) advance beyond simple attention by modeling both high- and low-frequency signals, they inherently decouple these components, thereby losing the correlated geometric structure between similarity and dissimilarity.
>
> DGAT introduces a fundamentally different geometric perspective: instead of treating neighborhoods as bags of independent signals, we model them as local manifolds where both parallel (similar) and orthogonal (dissimilar) components carry meaningful relational information. Our unified dual-attention mechanism preserves these geometric correlations by operating directly on embeddings rather than decomposing them into frequency bands.
>
> The key distinction lies in how we handle dissimilarity: while prior work typically treats it as noise to be removed or suppressed, DGAT models it as a geometric integral over the local manifold. This allows us to identify and integrate differences in a controlled manner through functional orthogonal and torsion gates, without enforcing artificial normalization constraints that force similarity and dissimilarity to sum to one.
>
> This geometric foundation increases the expressivity through destination-node regulation instead of focusing only on the neighbors’ embedding. Unlike methods that filter information at the source, DGAT ensures dissimilar information contributes meaningfully to the target embedding, leading to more stable propagation dynamics. We also make use of our adaptive scaling to control these dynamics. By allowing unnormalized, positive-valued components, DGAT adapts to neighborhood complexity rather than forcing artificial normalization. When the adaptation includes orthogonal information, we can also provide noise robustness. The manifolds perspective naturally handles noisy graphs where simple similarity measures break down.
>
> Most importantly, DGAT should be viewed as complementary rather than strictly superior: it provides a geometric substrate that can enhance existing architectures. Since it operates purely at the embedding level, DGAT can be integrated with frequency-based methods, contrastive approaches, or newer attention variants, potentially increasing their performance while providing geometric interpretability.
>
> Our empirical results demonstrate that this geometric approach particularly excels in noisy settings and on graphs with mixed homophily and heterophily, where is essential to preserve the correlation between nodes.
>
> ---
>
> **Q2**. To address this concern, we have incorporated additional modern graph Transformer baselines that are not simple extensions of classical attention mechanisms.
>
> Our initial selection focused on older models because many recent architectures primarily wrap standard attention layers with additional modules (e.g., positional encodings, embedding structure, global tokens), making it difficult to isolate improvements that come specifically from the attention mechanism itself. Since our goal is to evaluate DGAT at the attention level, using classical models we intended to ensure a fair and controlled comparison.
>
> But the concern that has arisen has led us to include some more recent graph transformers. Several recent transformers introduce fundamentally different aggregation schemes and therefore provide meaningful baselines. We have added (Tables 1 and 2 in the paper) Polynormer and Exphormer, which offer efficient or expressive alternatives to standard attention.
>
> - **Exphormer**: Sparse Graph Transformer (arXiv:2303.06147)
> - **Polynormer**: Polynomial-Expressive Graph Transformer (arXiv:2403.01232)
>
> ---
>
> **Q3**. Thank you for your suggestion and kind remark. We have conducted an ablation study. This study has revealed that the use of *w-orth* gate and torsion gate is flexible and dependent on the input dataset, as the authors of ACMGNN highlighted in their paper. We emphasize that instead of applying channel-wise attention conditioned on the adjacency matrix, we are adding or removing geometric gates. The performance for all the alternative model variants considered are presented in Table 15 in the appendix section A.3. All these variants were evaluated on every heterophilic dataset. Full details are provided in the paper.
>
> ---
>
> We sincerely thank the reviewer for their time and insightful comments. We hope that our responses will help clarify additional aspects of our work.

---

### Official Review · Reviewer_c2Dv · 2025-10-26

**Soundness:** 2
**Presentation:** 3
**Contribution:** 3
**Rating:** 4
**Confidence:** 3

**Summary:**

The manuscript introduces Difference-Based Graph Attention Networks (DGAT), a novel GNN architecture that extends standard attention mechanisms to jointly capture similarity and dissimilarity between nodes. The method integrates a dual attention pathway, which contains a similarity-based component using cosine or additive attention and a dissimilarity-based component using orthogonal projections and wedge-product approximations. The final representation is obtained via a learned gating mechanism that balances both components. DGAT is grounded in the Iwasawa–Cayley decomposition, offering geometric interpretability connecting orthogonal, scaling, and shifting operations. Experiments are conducted on multiple homophilic (OGBg-MolHIV, OGBn-Proteins, OGBl-PPA, OGBl-DDI) and heterophilic (Minesweeper, Roman-Empire, Amazon-Ratings, Questions) benchmarks, showing consistent performance improvements over GAT, GATv2, and Graph Transformer baselines

**Strengths:**

1. The proposed dual-path mechanism and gating function of DGAT address an underexplored limitation of GAT-like models that emphasize only similarity

2. DGAT based on the theoretical grounding via the Iwasawa–Cayley decomposition. This linkage provides interpretability and situates DGAT in a geometric-algebraic context.

3. The authors conduct comprehensive experiments, covering both homophilic and heterophilic benchmarks. The results show the improvements of DGAT compared with baselines.

**Weaknesses:**

1.The authors claim the convergence guarantees of DGAT in the introduction. However, I do not find the specific proof or lemma to quantify this property. If boundedness can lead to convergence, the author needs to clearly point this out and provide a proof.

2.Ablation studies are missing. For example, the selection of $\lambda$ and similarity v.s. dissimilarity isolation are not empirically separated.

3.Computational overhead and training stability. Claims about efficiency and non‑expansiveness lack runtime or convergence curves.

**Questions:**

1.In Section 1, the authors claim that DGAT is “non-expansive” with “convergence guarantees.” However, no formal proof, theorem, or empirical demonstration is included.

Question: Could you please provide the precise mathematical derivation or an outline of the argument that establishes non-expansiveness and convergence guarantees? Are these guarantees derived from the gating mechanism, the orthogonal projection, or both?

2.Eq. (4) introduces different versions of the gating function (i.e., w‑orthogonal, torsion, and default). The manuscript names them but provides no training dynamics or comparison.

Question: How do these gating variants quantitatively differ in learned behavior or performance? Have you tested their contributions through ablations, and if not, could you clarify why certain gates are only conceptually presented but not empirically analyzed?

3.I observe that the authors claim the efficiency related to head compensation and GPU-based orthogonal projection (in section A.5), yet no timing results are presented.

Question: What is the relative runtime cost of DGAT compared to GATv2 and Graph Transformer? Does the dissimilarity computation introduce additional latency or memory bottlenecks?

---

> ### Author Response · Authors · 2025-11-26
>
> We sincerely thank the reviewer for the kind remarks and the time dedicated to reviewing this paper. The **questions** raised highlight important aspects that were missing from the initial explanation. We address them one by one below.
>
> ---
>
> ### **Q1. Non-expansive and convergent behavior**
>
> We are grateful for the suggestion to include the mathematical derivation or an outline of the argument that establishes the non-expansive and convergent behavior of DGAT.
>
> A key challenge during message passing is preserving meaningful differences between nodes. In many existing methods (e.g., GARN, FAGCN, ACMGNN), dissimilarity information is often suppressed or removed. In contrast, DGAT explicitly aims to capture and diffuse non-common information without losing the shared information between nodes. This removes the “zero-sum” behavior, allowing controlled diffusion of information that lies between nodes’ probabilistic gates. The gating mechanism controls the diffusion, ensuring it does not grow excessively while maintaining orthogonal contributions. This is the main source of DGAT’s non-expansive behavior. The orthogonal projection ensures that the diffused information adds independent contributions to embeddings, preventing unbounded growth. The torsion gate removes combinations that are not suited for the problem at hand, which can be beneficial or neutral depending on the dataset. And last the dropout acts as a crucial stabilizer, connecting these components and helping ensure that learned information is robust.
>
> References:
>
> * [DOI:10.1016/j.neunet.2024.106484] Graph Aggregating-Repelling Network (GARN) (https://doi.org/10.1016/j.neunet.2024.106484)
> * [arXiv:2210.07606] Revisiting Heterophily For Graph Neural Networks (https://arxiv.org/abs/2210.07606)
> * [arXiv:2101.00797] Beyond Low-frequency Information in Graph Convolutional Networks  (https://arxiv.org/abs/2101.00797)
>
> Regarding convergence, if the model is trained correctly, the gating parameter (λ) can augment, reduce, or suppress dissimilarity propagation, effectively reverting the model to a standard GATv2 or Graph Transformer when needed. This ensures that, in the worst case, DGAT can self-stabilize during training and possesses mechanisms to control divergence. Due to space limitations, we could not add the mathematical proof to this response. We have added the proof to the manuscript, in the Appendix section A.3.1, in accordance with the reviewer’s helpful suggestion.
>
> ---
>
> ### **Q2. Ablation**
>
> Thank you for pointing out this missing part of our paper. We have added the results of the ablation study to Table 15 in the appendix section A.3. The table is organized into two parts, corresponding to the two used baselines GATv2 and Graph Transformer for DGATv2 and DGAT-transformer variants, respectively. The rows display the different combinations of the use of similarity, dissimilarity and gating. The results show that each dataset is very dependent on the use of a certain gate or not. Most of the dataset benefits of the usage of the w-orth gate, while the inner orthogonality may be beneficial in scenarios that require increased control over the geometry of the resulting embeddings. Some extra ablations in the future might test the use of non linear or non continuous functions such as splines or even non differentiable functions.
>
> ---
>
> ### **Q3. Runtime costs**
>
> We thank the reviewer for giving us the opportunity to clarify this point. We have compared the execution times of our models with baselines of different types on the eight datasets considered. A summary of this comparison is presented in the paper’s appendix section A.4.
>
> All DGAT variants (DGATv2, DGAT-transf and DGAT-transf-gram) are somewhat less efficient, to varying degrees, than the Linear type attention Transformer models, with the exception of Nodeformer, which exhibits execution times similar to —and sometimes even higher than— those of the DGAT variants.
>
> However, several points should be highlighted. First, in the comparison with Graph Transformer, the percentage increase in execution time is very small for the datasets with higher computational cost, and the larger percentage increases occur mainly in the lower-cost datasets, where they are less impactful. Second, in the comparison with GATv2, the difference in execution time is larger, although DGATv2 consistently outperforms GATv2 in all datasets (as shown in Table 2 of the paper). Third, we would also like to point out that DGAT variants are not yet optimized, and future work could further improve its efficiency. Therefore, it can be argued that the performance gain can outweigh the additional latency, given that no memory bottlenecks were observed.
>
> ---
>
> Once again, we would like to thank the reviewer for his/her valuable time and constructive feedback.
> We believe that the revisions based on these comments have substantially improved the quality and clarity of the manuscript.

---

> > ### Comment · Reviewer_c2Dv · 2025-11-28
> >
> > Has the author provided the latest version of the revised manuscript? The proof for section A.3.1 does indeed appear to be missing.

---

> ### Author Response · Authors · 2025-11-28
>
> We sincerely apologize for the mistake. The correct section for the mathematical proof is the A.2.1 in the paper.

---

### Official Review · Reviewer_eYdE · 2025-11-02

**Soundness:** 3
**Presentation:** 3
**Contribution:** 3
**Rating:** 6
**Confidence:** 3

**Summary:**

The paper proposes DGAT, a dual-path attention layer that couples a standard similarity path with a dissimilarity/orthogonal path computed via Gram-projection / wedge-product surrogates and combined with a learned gate (including an orthogonality-enforcing “w-orth” and an optional torsion/Lie-bracket-inspired variant). Core updates are given in Eqs. (1)–(3) and the gate in Eq. (4).

**Strengths:**

1. Clear, principled idea. The orthogonal/difference channel complements similarity attention to encode contrast rather than only alignment; the equations and gate make the mechanism explicit and modular.
2. Geometric grounding. The Iwasawa perspective provides an interpretable decomposition (K/A/N) mapping cleanly onto (similarity / difference / gating). This is rare in GNN attention papers and helps motivate design choices.
3. Empirical signal. On OGB tasks and heterophilic benchmarks, DGAT variants are reported to outperform GAT/GATv2/Graph Transformers under matched settings (with a head-count compensation to control params).

**Weaknesses:**

1. While the geometric story is appealing, some claims (e.g., “non-expansive and convergent operator”) are mentioned in the intro but I did not see full proofs in the provided snippets; if they exist in the appendix, make them crisp with assumptions and operator norms (Lipschitz constants, spectral bounds). (Pointer to tighten: Sections A.3–A.4 already frame energy/orthogonality—turn these into formal theorems.)
2. The paper introduces w-orth and torsion gates; more ablation would clarify when each helps, sensitivity to λ, and whether improvements persist if gates are simplified. (Some hyperparameter tables appear, but targeted gate ablations would strengthen claims.)
3. It seems that results are described as best-of-run with std as “difference between best results,” which is non-standard. Prefer mean±std over many seeds.

**Questions:**

1. Can the w-orth constraint hurt when neighborhoods are tiny/noisy?

---

> ### Author Response · Authors · 2025-11-22
>
> We sincerely thank you for your time and all your valuable feedback and helpful comments.
> We really appreciate your **question**. Your question kindly remarks the weak and important points of the orthogonality this model includes.
>
> The w-orthogonality gate is inherently a double-edged mechanism. On one hand, it is highly effective at suppressing spurious or irrelevant signals; on the other hand, in settings where neighborhoods are extremely small or dominated by noise, the same mechanism may inadvertently amplify undesirable variations. Nevertheless, the advantage of the w-orth gate lies in its geometric structuring of noise: any residual information is forced to align with an orthogonal subspace defined from the perspective of the source node. This transformation makes the noise informative in the sense that potentially meaningful but weak signals become isolated, enabling the optimizer to more precisely distinguish between informative components and actual noise.
>
> Therefore, in response to the question: yes, the *w-orth* constraint can both help and hurt, depending on the data and training dynamics. Empirical results indicate that it is particularly beneficial in scenarios where nodes require strong feature separation or where discriminative signals are inherently weak. In such cases, the gate’s effect is complemented by the model's internal orthogonalization dynamics, which help maintain robustness even under sparsity or local heterogeneity. This explanation is included at the end of A.2.4 in the appendix.
>
> Below, we also address the issues you mentioned as **weaknesses**.
>
> ---
>
> 1. We are grateful for the suggestion about the proof of non-expansive and convergent behavior.
> During message passing, a key challenge is preserving meaningful differences between nodes. In many existing methods (e.g., GARN , FAGCN , ACMGNN ), dissimilarity information is often suppressed or removed. In contrast, DGAT explicitly aims to capture and diffuse non-common information without losing the shared information between nodes. This removes the “zero-sum” behavior, allowing controlled diffusion of information that lies between nodes’ probabilistic gates. The gating mechanism controls the diffusion, ensuring it does not grow excessively while maintaining orthogonal contributions. This is the main source of DGAT’s non-expansive behavior. The orthogonal projection ensures that the diffused information adds independent contributions to embeddings, preventing unbounded growth. The torsion gate removes combinations that are not suited for the problem at hand, which can be beneficial or neutral depending on the dataset. And last the dropout acts as a crucial stabilizer, connecting these components and helping ensure that learned information is robust.
> Regarding convergence, if the model is trained correctly, the gating parameter (λ) can augment, reduce, or suppress dissimilarity propagation, effectively reverting the model to a standard GATv2 or Graph Transformer when needed. This ensures that, in the worst case, DGAT can self-stabilize during training and possesses mechanisms to control divergence. Due to space limitations, we could not add the mathematical proof in the response. It has been added to the paper and can be found in the Annex section A.2.1.
>
> ---
>
> 2. Thank you for your valuable comment on ablation. We agree that the omission of the ablation study is a weak point of our paper and that addressing this point improves clarity. Following this advice, we have included the ablation study performed. Due to space limitations, it cannot be added to this response, but it has been included in the paper in section A.3.
>
> ---
>
> 3. Thank you for highlighting this point about the standard for the performance results. Following your helpful suggestion, we have changed the way we get the evaluation results. We have adopted the evaluation procedure followed in other papers, for example (there are many others):
> * [arXiv:2406.08993] Classic GNNs are Strong Baselines: Reassessing GNNs for Node Classification (https://arxiv.org/abs/2406.08993)
> * [arXiv:2210.07606] Revisiting Heterophily For Graph Neural Networks (https://arxiv.org/abs/2210.07606)
> * [arXiv:2403.01232] Polynormer: Polynomial-Expressive Graph Transformer in Linear Time (https://arxiv.org/abs/2403.01232)
>
> The change made is taking the highest metric (Accuracy, AUROC,..) of each  training and making the mean with the highest values of independent trainings. Each training makes use of different seeds. The revised version of the paper includes the values, using those works as the basis for the experimental setup.
>
> ---
>
>
> We hope that our responses, along with the revisions incorporated into the manuscript, have clarified the issues raised. We are sincerely grateful for the reviewer’s constructive and insightful comments.

---

### Author Response · Authors · 2025-12-03
**Summary of the discussion process**

In light of the situation, and to facilitate the review process, we summarize the key aspects of our work, the reviewers’ feedback (strengths and revision requests), and the actions taken to address each request.

The key contribution of our work is the insight that dissimilarity, instead of being treated as noise and discard it, can provide valuable information in certain contexts. Consequently, **the attention mechanisms should be designed to exploit dissimilarity** accordingly.
We conducted an extensive study on the benefits of extracting information from the dissimilarity, using orthogonality to perform this extraction. The study considered widely used benchmarks for both homophilic and heterophilic graphs and compared the baselines with 3 variants of our model (DGAT).

Our empirical results demonstrate that this geometric approach particularly excels in noisy settings and on graphs with mixed homophily, where is essential to preserve the correlation between nodes. Therefore, the research emphasizes the relevance of the proposed idea. Rather than claiming universal superiority, we position our approach as a principled framework that addresses fundamental limitations in how graph attention models relational geometry. We also want to highlight that our approach is complementary to other existing approaches.

**The reviewers’ feedback is highly favorable with regard to the strengths of our work**. All reviewers highlight the **novelty** of the idea and the **new insights** it brings and emphasize the **strong theoretical grounding and interpretability**, offered through the Iwasawa–Cayley decomposition. Two of the reviewers also highlight the extensive experimental evaluation.

Concerning the reviewers’ questions and suggestions:

-	The reviewers request for information that was not included in the first version of the paper — namely, the study of ablation and the mathematical proof of the claimed “non-expansive and convergent” behavior. We fully agree that both aspects were missing and they have now been incorporated.

-	Reviewers 21ta and Jp3v suggested additional methods that, in their opinion, should be considered as baselines (more recent graph Transformers for one of them, and GARN, GCN, GraphSAGE, and JKNet for the other). We have explained why these baselines were not initially included, and, following their suggestion, we have clarified the details and added them to our study. Specifically, as our model introduces a modification to the attention mechanism, it should be compared against a model that uses the same mechanism in its unmodified form. Regarding GARN, no public implementation is available. Consequently, we have implemented it ourselves to include it as suggested.

-	Reviewers 21ta and Jp3v raised questions regarding the research gap and the motivation of our work. The first one explicitly questioned whether other methods such as FAGCN and ACMGN could solve the problem addressed, since they capture both low-frequency information (similarity) and high-frequency information (dissimilarity). **In our responses, we have clarified that the contribution of our model is to leverage dissimilarity** rather than simply remove it (as previous methods do), extracting useful information from it. **The main goal of our study is to address a limitation in attention-based graph models** (such as GARN, FAGCN, ACMGCN, and A2GCN), **namely their limited use of node dissimilarity**.

-	Four additional specific questions or suggestions have also been addressed and incorporated into the manuscript.

The table below summarizes the weaknesses (W) and questions (Q) raised by the reviewers (in columns), along with our actions and the modified sections of the paper.
| | eYdE | c2Dv | 21ta | Jp3V |
|-|-|-|-|-|
| Non-expansive and Convergent. Included (Appendix A.2.1) | W1 | Q1 |  |  |
| Ablation. Included (Appendix A.3) | W2 | Q2 | Q3 | |
| Baselines suggested. Added (results in Paper Tables 1 and 2) |  |  | Q2. Added Polynormer and Exphormer | W4. Added GCN, GraphSage, GARN, JKNet, and ResNet |
| Research gap / Motivation |  |  | Q1: Gap is overclaimed. Explained and added FAGCN as baseline (results in Paper Tables 1 and 2) | W2: Difference/relation to graph contrastive learning. W3: The motivation is unclear. Clarified and included (Paper, Section 3.1) |
| Other questions | Q1: Concern about tiny/noisy neighborhoods. Answered and included (Appendix A.2.4) | Q3: Runtime cost. Added (Appendix A.4) | | |
| Other concerns | W3: Standard for results. Changed (Paper, Tables 1 and 2) | | | W1: Diagram/illustration. Added (Paper, Figure 1) |

In summary, we have carefully addressed all questions and incorporated all suggestions raised during the review process, and we are confident that our responses adequately clarify every point that emerged. We therefore expected a high degree of satisfaction from the reviewers, and we believe that our score would have improved substantially with this revised version of our paper.

---

### Meta-Review · Area_Chair_zsfn · 2025-12-22

**Summary:**

The reviewers have raised the following major concerns:

(1) Incomplete theoretical analysis.

(2) Missing ablation studies.

(3) Missing benchmarks.

(4) Missing complexity analysis.

(5) Motivation regarding considering feature similarity.

(6) Detailed analysis of the model for homophilic and heterophilic datasets.

(7) Comparing with graph contrastive learning.

(8) Lacks an intuitive diagram.

**Reviewer Concerns:**

The authors have provided comparisons with (suggested benchmarks) such as GARN, FAGCN. In my opinion, this is sufficient, given the limited rebuttal period. Moreover, ablation studies are performed (Table 15), and run-time is included (Table 16). An intuitive diagram is provided in Fig. 1.

However, some of the conceptual discussions would benefit from further clarification and strengthening. For example, the theoretical discussion in Appendix A.2.1 is not very informative in its current form and lacks sufficient rigor, making it difficult to see how the results support the claim that the proposed framework is expected to perform well. From a mathematical perspective, the underlying setup on manifolds and fiber bundles does not appear to be clearly or fully defined. In particular, Corollary 1.1 is not stated in a precise mathematical manner, which makes its interpretation and validity unclear. Additionally, the connection to Iwasawa decomposition and Lie theory seems rather high-level, and a more rigorous and explicit mathematical correspondence would help clarify the relevance of these concepts to the proposed framework.

The newly added comparisons, ablation studies, and complexity analysis indicate that incorporating dissimilarity yields only marginal performance gains, while noticeably increasing computational cost. This suggests that a clearer justification of the trade-off between performance improvement and added complexity would strengthen the contribution.

Some of the conceptual concerns raised by Reviewer Jp3V (e.g., regarding motivation and heterophily) are well taken and warrant deeper consideration. These points are not yet sufficiently addressed in the current rebuttal.

**Reviewer Scores:**

None of the reviewers has actively participated in the discussion. I think reviewer 21ta might increase the score from 2 to 4, while the remaining reviewers will maintain their scores.

---

### Decision · Program_Chairs · 2026-01-26

Reject